# CRK2 controls cytoskeleton morphogenesis in *Trypanosoma brucei* by phosphorylating β-tubulin to regulate microtubule dynamics

**Kyu Joon Lee, Qing Zhou, Ziyin Li** *

Department of Microbiology and Molecular Genetics, McGovern Medical School, University of Texas Health Science Center at Houston, Houston, Texas, United States of America

* Ziyin.Li@uth.tmc.edu

## Abstract

Microtubules constitute a vital part of the cytoskeleton in eukaryotes by mediating cell morphogenesis, cell motility, cell division, and intracellular transport. The cytoskeleton of the parasite *Trypanosoma brucei* contains an array of subpellicular microtubules with their plus-ends positioned toward the posterior cell tip, where extensive microtubule growth and cytoskeleton remodeling take place during early cell cycle stages. However, the control mechanism underlying microtubule dynamics at the posterior cell tip remains elusive. Here, we report that the S-phase cyclin-dependent kinase-cyclin complex CRK2-CYC13 in *T. brucei* regulates microtubule dynamics by phosphorylating β-tubulin on multiple evolutionarily conserved serine and threonine residues to inhibit its incorporation into cytoskeletal microtubules and promote its degradation in the cytosol. Consequently, knockdown of CRK2 or CYC13 causes excessive microtubule extension and loss of microtubule convergence at the posterior cell tip, leading to cytoskeleton elongation and branching. These findings uncover a control mechanism for cytoskeletal microtubule dynamics by which CRK2 phosphorylates β-tubulin and fine-tunes cellular β-tubulin protein abundance to restrict excess microtubule extension for the maintenance of cytoskeleton architecture.

## Author summary

*Trypanosoma brucei* is a protozoan parasite causing human and animal trypanosomiasis in sub-Saharan Africa. This parasite has a cytoskeleton composed of an array of subpellicular microtubules, which maintains cell morphology and cell division. The microtubules located at the posterior tip of the parasite are highly dynamic during early cell cycle stages, but the underlying mechanism remains elusive. Here, we found that the S-phase cyclin-dependent kinase-cyclin complex CRK2-CYC13 regulates microtubule dynamics by phosphorylating β-tubulin, a subunit of the microtubules, to inhibit β-tubulin incorporation into cytoskeletal microtubules and promote its degradation in the cytoplasm. As a consequence, knockdown of CRK2 or CYC13 causes excessive microtubule extension at the posterior cell tip, resulting in posterior elongation and branching. These results reveal a

**Data Availability Statement:** All data are in the manuscript and/or supporting information files.

**Funding:** This work was supported by the National Institutes of Health R01 grants AI118736 and

AI101437 to Z.L. The funders had no role in study design, data collection and analysis, decision to publish, or preparation of the manuscript.

**Competing interests:** The authors declare that they have no conflicts of interest with the contents of this article.

critical role of CRK2-CYC13 in maintaining cytoskeleton morphology by fine-tuning β-tubulin protein abundance to restrict excess microtubule extension.

## Introduction

Microtubules are dynamic cytoskeletal filaments composed of heterodimers of α- and β-tubulin proteins, and are involved in diverse cellular functions, such as cell morphogenesis, cell motility, cell division, and intracellular transport in eukaryotes. The functional diversity of the structurally conserved microtubules is attributed to the distinct microtubule identities generated by the expression of different tubulin isoforms as well as post-translational modifications (PTMs) [1]. Both the α- and β-tubulin proteins are subjected to a wide range of PTMs, which alter the dynamics, the organization, and the interaction of microtubules with other cellular components [2]. Some of the PTMs, such as acetylation, phosphorylation, and palmitoylation, are ubiquitous, whereas some other PTMs, such as tyrosination, polyglutamylation, and poly-glycylation, are likely unique to tubulin [1]. Different types of PTMs on the microtubules generate signals with different information complexity, and the locations of the PTMs on tubulin affect different functions of microtubules [1]. Therefore, post-translational modifications of tubulin proteins generate a regulatory mechanism for controlling the specialization of the microtubule cytoskeleton.

*Trypanosoma brucei* is a flagellated, unicellular parasite causing sleeping sickness in humans and nagana in cattle in sub-Saharan Africa. The cytoskeleton of *T. brucei* contains a corset of subpellicular microtubules, and it plays crucial roles in maintaining cell shape and intracellular organelle positions [3,4]. The subpellicular microtubules form a helical cage-like structure along the longitudinal cell axis, with their plus-ends placed at the posterior end of the cell [5]. During the *T. brucei* cell division cycle, the microtubule cytoskeleton is duplicated through a templated biogenesis of the subpellicular microtubule array for a semi-conservative distribution to the two daughter cells [6,7]. From early cell cycle stages, newly assembled microtubules are inserted into the subpellicular microtubule array between the existing microtubules, and the corset microtubules start to elongate at the posterior cell tip [8]. During post-mitotic phases of the cell cycle, some very short microtubules are nucleated laterally to intercalate the microtubules in the subpellicular microtubule array [8]. Following cytokinesis that occurs along the longitudinal cell axis, the two daughter cells each inherit a complete subpellicular microtubule corset, although extensive post-cytokinesis microtubule remodeling occurs at the anterior tip of the new-flagellum daughter cell and at the posterior tip of the old-flagellum daughter cell [9]. *T. brucei* also possesses a microtubule-based axoneme in its motile flagellum, a subset of quartet microtubules associated with the flagellum attachment zone (FAZ) filament, and a subset of intra-nuclear microtubules dedicated to the assembly of a mitotic spindle. The axonemal microtubules have an opposite polarity to the subpellicular microtubules, with their plus ends directed toward the anterior of the cell [3]. The polarity of the quartet microtubules has not been determined, but their plus ends are assumed to be located at the anterior tip of the cell.

Clusters of alternating α- and β-tubulin gene repeats are present in the *T. brucei* genome, and all copies of the respective tubulins have identical sequences [10,11]. Due to the polycistronic transcription of genes in *T. brucei*, this unique organization of tubulin genes likely enables co-expression of the microtubule-building unit proteins at levels sufficient to support the assembly and maintenance of the subpellicular microtubule array, the flagellar axoneme, and the FAZ-associated microtubule quartet. Microtubule PTMs, including acetylation [12],

tyrosination [13], detyrosination [14], phosphorylation [15], and polyglutamylation [16–18] have been detected in *T. brucei*. Among these PTMs, the polyglutamylation and detyrosination of microtubules have been functionally characterized [14,17,18], both of which are required for maintaining cytoskeletal architecture at the cell posterior. The function of other forms of PTMs on microtubules in *T. brucei* remains elusive and merits further exploration.

Microtubule extension at the posterior portion of trypanosome cells appears to be regulated to coordinate with cell differentiation or cell cycle progression. Overexpression of a CCCH zinc-finger protein, TbZFP2, in the procyclic form of *T. brucei* causes the elongation of the cell posterior due to polar extension of microtubules, which was termed "nozzle" phenotype [19]. Knockdown of the G1 cyclins CYC2 [20,21] or CYC7 [22], knockdown of the G1 cyclin-dependent kinase homolog CRK1 [22], and double knockdown of CRK1 and CRK2 [23,24] in the procyclic form compromise the posterior morphology of G1-arrested cells, resulting in posterior elongation and, occasionally, posterior branching [20–23]. Knockdown of the Sphingosine Kinase homolog in *T. brucei*, TbSPHK, also arrests cells at the G1/S boundary and causes posterior elongation [25]. A recent study of microtubule polyglutamylation in *T. brucei* showed that knockdown of tubulin polyglutamylase genes causes posterior elongation and branching, which was termed "glove" phenotype [18], suggesting the requirement of tubulin polyglutamylation for maintaining posterior cytoskeleton architecture in *T. brucei*. Proper formation of the posterior cell tip in *T. brucei* also requires two microtubule-associated proteins named PAVE1 and PAVE2, which localize to the posterior and the ventral side of the cell body and stabilize the subpellicular array microtubules at the cell posterior [26,27]. Knockdown of PAVE1 or PAVE2 disrupts the formation of the tapered posterior end, although the cell posterior is not evidently elongated [26,27].

We recently demonstrated that CRK2 in *T. brucei* functions as an S-phase cyclin-dependent kinase and associates with a new cyclin named CYC13 [28]. Knockdown of CRK2 or CYC13 in the procyclic form of *T. brucei* causes defective S-phase progression and appears to also produce cells with an elongated posterior or an elongated and branched posterior [28]. The CRK2-CYC13 complex promotes DNA replication by phosphorylating Mcm3 to facilitate the assembly of the DNA replicative helicase complex, the Cdc45-Mcm2-7-GINS complex [28], but its potential role in posterior cytoskeleton morphogenesis was not explored in the previous study. Here, we investigated the effects of RNAi of CRK2 or CYC13 on posterior morphogenesis and showed that the CRK2-CYC13 complex was required for maintaining posterior cytoskeleton morphology by restricting excess microtubule extension and promoting microtubule convergence at the posterior cell tip. We further identified β-tubulin as an *in vitro* substrate of CRK2 and identified multiple CRK2 phosphosites, including three evolutionarily conserved residues Ser-172, Ser-351, and Thr-372. Finally, we demonstrated that CRK2-mediated phosphorylation of β-tubulin impaired its incorporation into microtubules and promoted its degradation in the cytosol. These findings uncovered a new role for the S-phase cyclin-dependent kinase in regulating microtubule dynamics to control cytoskeleton morphogenesis in this early branching eukaryote.

## Results

### The CRK2-CYC13 complex is required for posterior cytoskeleton morphogenesis in *T. brucei*

In addition to arresting cells at the S-phase of the cell cycle in the procyclic form of *T. brucei* [28], RNAi of CYC13 and RNAi of CRK2 both caused posterior elongation and branching (Fig 1A). The posterior of a procyclic trypanosome cell, which we defined here as the part of the cell body from the kinetoplast to the posterior cell tip, is known to extend through the

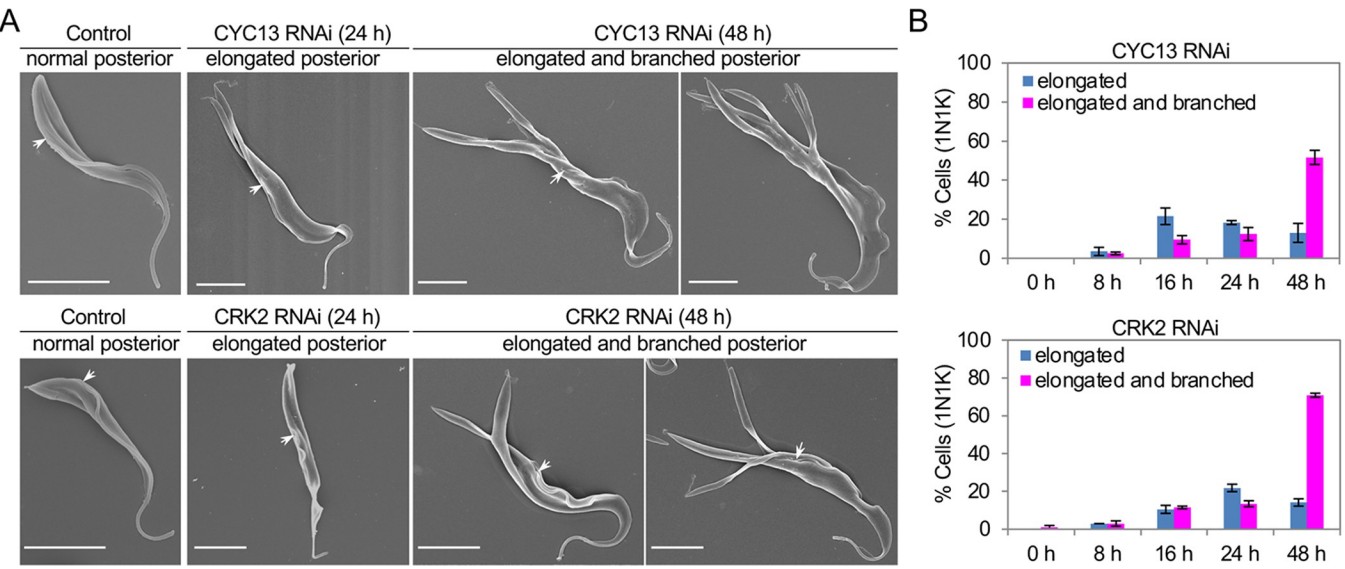

**Fig 1. Knockdown of CYC13 or CRK2 causes posterior elongation and branching in the procyclic form of *T. brucei*.** (**A**). Scanning electron microscopic analysis of non-induced control, CYC13 RNAi-induced cells, and CRK2 RNAi-induced cells. The white arrows indicate the position of the flagellum pocket. Scale bars: 5 μm. (**B**). Quantitation of cells with an elongated posterior or an elongated and branched posterior of non-induced and RNAi-induced cells of the CYC13 RNAi cell line and the CRK2 RNAi cell line. 100 1N1K cells for each time point were used for measurement of the posterior length (K-to-P) and counted. Error bars indicate S.D. from three independent biological replicates.

growth of the corset microtubules during early cell cycle stages [8]. We measured the length of the posterior of 1N1K (one nucleus and one kinetoplast) cells of the non-induced control and the two RNAi cell lines after RNAi induction for up to 48 hours, and we classified the elongated posterior of the RNAi cells as being longer than the posterior of any control 1N1K cells. These analyses showed that the 1N1K cells with an elongated posterior emerged after 8 hours of RNAi induction for both RNAi cell lines, peaked at ~21% after 16 h for CYC13 RNAi and at ~22% after 24 h for CRK2 RNAi, and then decreased (Fig 1B). The 1N1K cells with a branched posterior, which was also elongated, emerged after 8 hours of RNAi and increased to ~52% after 48 h for CYC13 RNAi and ~71% after 48 h for CRK2 RNAi (Fig 1B). It appears that after RNAi of CRK2 or CYC13, the cell posterior was first elongated and then branched out to form two to more than five branches (Fig 1A).

To quantitatively analyze the effect of CYC13 RNAi and CRK2 RNAi on cell morphology, we measured the overall cell body length, i.e., the distance between the posterior (P) and the anterior (A) of the cell, and the position of the nucleus (N) and the kinetoplast (K) relative to the posterior and the anterior (Fig 2A). RNAi of CYC13 and RNAi of CRK2 both caused an increase in the cell body length (P-to-A) from an average of ~18.4 μm to ~29.8 μm and ~30.2 μm, respectively (Fig 2B). For the kinetoplast, its distance to the posterior (K-to-P), to the anterior (K-to-A), and to the nucleus (K-to-N) was increased from an average of ~4.6 μm to ~9.1 μm and ~9.4 μm, from an average of ~12.5 μm to ~20.8 μm and ~21.1 μm, and from an average of ~3.2 μm to ~6.9 μm and ~6.9 μm, for CYC13 RNAi and CRK2 RNAi, respectively (Fig 2B). For the nucleus, its distance to the posterior (N-to-P) was increased from an average of ~8.2 μm to ~16.2 μm and ~16.3 μm for CYC13 RNAi and CRK2 RNAi, respectively (Fig 2B), whereas the distance from the nucleus to the anterior (N-to-A) was only slightly (but insignificantly) increased from an average of ~11.1 μm to ~13.1 μm and ~12.5 μm for CYC13 RNAi and CRK2 RNAi, respectively (Fig 2B), indicating that the location of the nucleus likely remained unchanged. These results suggest that RNAi of CYC13 or CRK2 caused an excessive

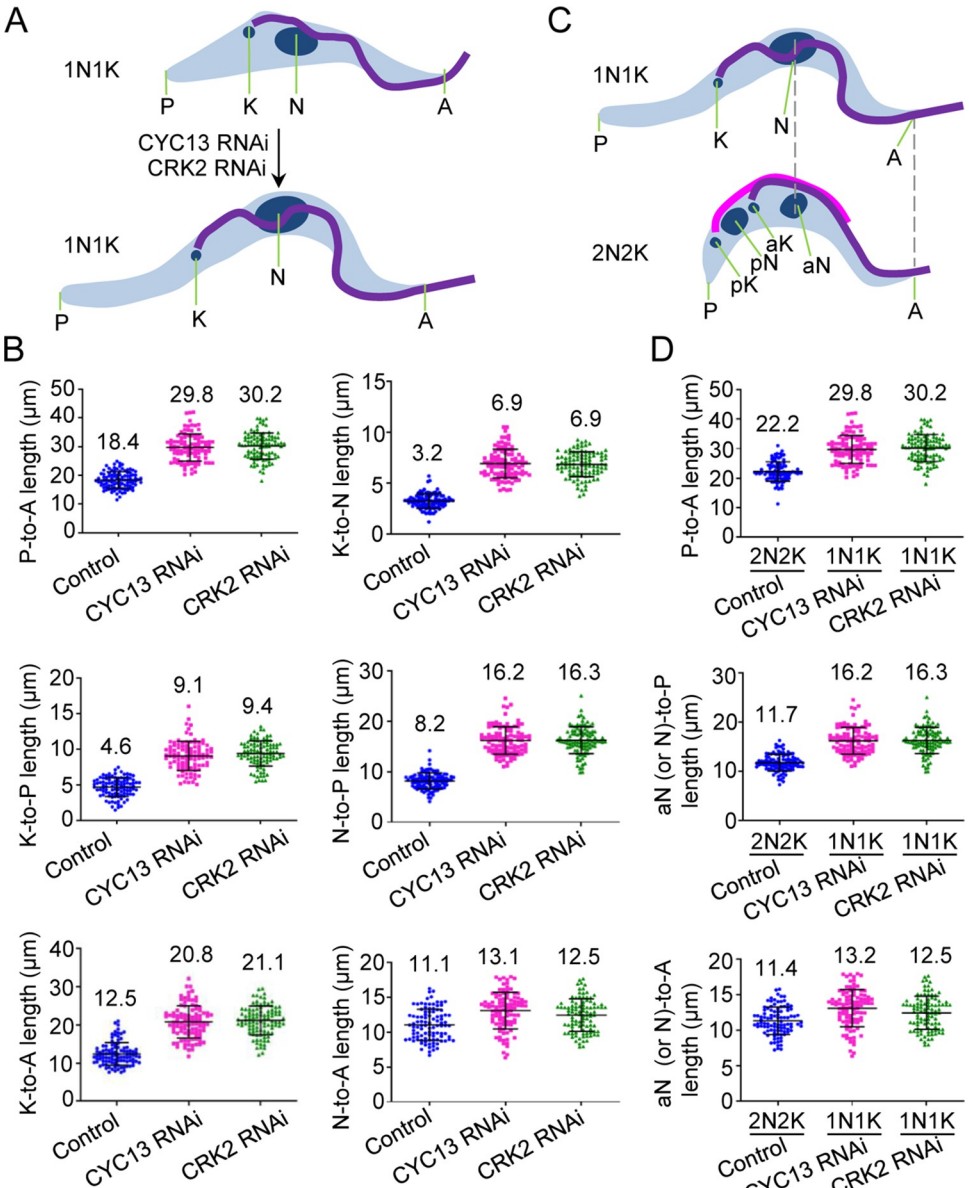

**Fig 2. Analysis of the effects of CYC13 RNAi and CRK2 RNAi on cell morphology.** (**A**). Schematic drawing of 1N1K cells from non-induced control and RNAi of CYC13 or CRK2, showing the effect of RNAi on cell morphology, for the measurement presented in panel **B**. P, posterior; A, anterior; K, kinetoplast; N, nucleus. (**B**). Measurement of the distance between kinetoplast, nucleus, cell posterior, and cell anterior of the 1N1K cells from non-induced control, CYC13 RNAi (48 h), and CRK2 RNAi (48 h). 100 cells from each cell line were used for measurement. The number on each plot indicates the average length. (**C**). Schematic drawing of a 1N1K cell from CRK2 RNAi or CYC13 RNAi and a 2N2K cell from the non-induced control for the measurement presented in panel **D**. P, posterior; A, anterior; K, kinetoplast; N, nucleus; pK, posterior kinetoplast; pN, posterior nucleus; aK, anterior kinetoplast; aN, anterior nucleus. (**D**). Measurement of the distance between cell posterior, cell anterior, and nucleus (1N1K cells) or anterior nucleus (2N2K cells) of the 2N2K cells from the non-induced control and the 1N1K cells after CYC13 RNAi (48 h) and CRK2 RNAi (48 h). 100 cells for each cell line were used for measurement. The number on each plot indicates the average length.

posterior elongation, leading to the increase of the cell body length and the migration of the kinetoplast, but not the nucleus, toward the posterior portion of the cell.

It was previously postulated that the growth of the posterior cytoskeleton through microtubule extension during the cell cycle is coupled to the G1/S transition in *T. brucei* [23]. In this regard, the observed "excessive" posterior elongation in CYC13 RNAi cells and CRK2 RNAi cells is attributed to the decoupling of the nuclear cycle from the process of posterior microtubule extension. In such a scenario, the cell body length of the 1N1K cells after CYC13 RNAi or CRK2 RNAi should not exceed the cell body length of wild-type cells at their maximal length during the entire cell cycle, i.e., the bi-nucleated (2N2K, two nuclei and two kinetoplasts) cells at late cell cycle stages. To test this possibility, we measured the cell body length of the 2N2K cells from the non-induced control population, and compared it with the cell body length of the 1N1K cells collected after CYC13 RNAi and CRK2 RNAi (Fig 2C). The results showed that the average cell body length (~22.2 μm) of the control 2N2K cells was substantially shorter than the average cell body length (~29.8 μm and ~30.2 μm, respectively) of the CYC13-deficient and CRK2-deficient 1N1K cells (Fig 2D). Additionally, since the nucleus of a 1N1K cell may remain at the same location, herein referred to as anterior nucleus (aN), during cell cycle progression to become a 2N2K cell (Fig 2C), then if the decoupling occurred, the average N-to-P distance in the CYC13-deficient and CRK2-deficient 1N1K cells should be similar to the aN-to-P distance in the control 2N2K cells. However, the average N-to-P distance (~16.2 μm and ~16.3 μm, respectively) in the CYC13-deficient and CRK2-deficient 1N1K cells was substantially longer than the aN-to-P distance (~11.7 μm) in the control 2N2K cells (Fig 2D). In contrast, the average N-to-A distance (~13.2 μm and ~12.5 μm, respectively) in the CYC13-deficient and CRK2-deficient 1N1K cells was not significantly different from the aN-to-A distance (~11.1 μm) in the control 2N2K cells (Fig 2D), indicating that the nucleus remained at the same location after CRK2 RNAi or CYC13 RNAi. Taken together, these results suggest that the observed elongated posterior of the CYC13-deficient and CRK2-deficient 1N1K cells was not a result of the decoupling of the nuclear cycle from the normal posterior elongation process during the cell cycle, but rather was due to the excessive growth of the posterior cytoskeleton.

We also observed that the elongated posterior of CYC13 RNAi cells and CRK2 RNAi cells assumed a blunt end, in contrast to the tapered posterior end in the non-induced control cells (Fig 3A), suggesting that microtubule remodeling at the posterior cell tip was likely impaired. To test this hypothesis, we prepared detergent-extracted cytoskeletons of trypanosome cells and examined the corset microtubules by transmission electron microscopy. In the control cells with a tapered posterior tip, the microtubule plus ends were converged together to form a pointed tip (Fig 3B); however, in CRK2 RNAi cells and CYC13 RNAi cells with a blunt posterior end, the microtubule plus ends apparently were not converged together (Fig 3B), suggesting that microtubule convergence at these locations was impaired in CYC13 RNAi cells and CRK2 RNAi cells.

CRK2 RNAi and CYC13 RNAi also appeared to additionally cause an increase in the size of the cell body and the length of the flagellum and its associated FAZ (S1 Fig). Quantitative analyses showed an increase in the cell body size of the 1N1K cells from an average of ~43 $\mu m^2$ to ~72 $\mu m^2$ after 24 h and to ~110 $\mu m^2$ after 48 h of CYC13 RNAi, and from an average of ~40 $\mu m^2$ to ~52 $\mu m^2$ after 24 h and to ~108 $\mu m^2$ after 48 h of CRK2 RNAi (S1A and S1B Fig). The length of the flagellum of the 1N1K cells was increased from an average of ~17 μm to ~25 μm after CYC13 RNAi for 48 h and to ~29 μm after CRK2 RNAi for 48 h (S1C and S1D Fig). Similarly, the length of the FAZ of the 1N1K cells was increased from an average of ~15 μm to ~21 μm after CYC13 RNAi for 48 h and to ~24 μm after CRK2 RNAi for 48 h (S1C and S1D Fig). These observations suggest that the CRK2-CYC13 complex is required to restrict over-expansion of the cytoskeleton, the flagellum, and the FAZ structure. CRK2 is likely to be involved in the regulation of microtubule dynamics of the subpellicular array, the flagellar axoneme, and the FAZ-associated microtubule quartet.

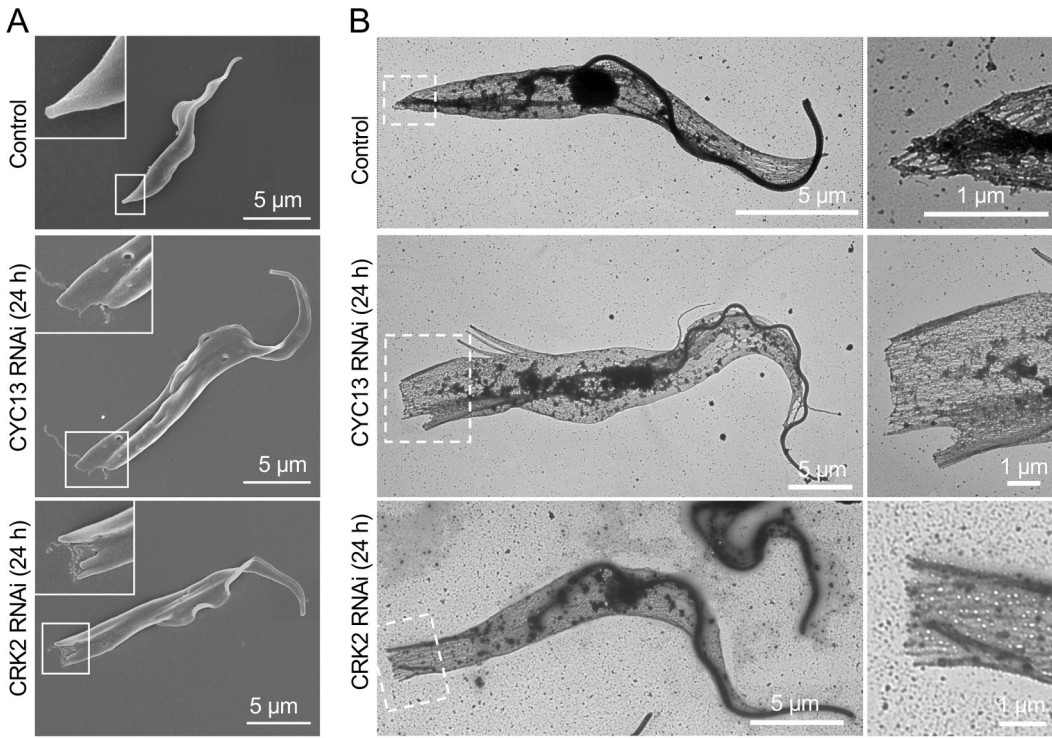

**Fig 3. Knockdown of CYC13 or CRK2 impairs posterior microtubule convergence.** (**A**). Scanning electron microscopic analysis of non-induced control cells, CYC13 RNAi cells, and CRK2 RNAi cells. Scale bars: 5 μm. (**B**). Transmission electron microscopic (TEM) analysis of the subpellicular microtubules at the cell posterior. Shown are TEM images of negatively stained cytoskeletons of non-induced control cells, CYC13 RNAi cells, and CRK2 RNAi cells. The right panels show the enlarged images of the boxed areas in the respective left panels. Scale bars: 5 μm and 1 μm as indicated.

## The CRK2-CYC13 complex restricts excess microtubule extension at the cell posterior

The elongation and branching of the posterior of the CYC13-deficient and CRK2-deficient cells suggests excessive microtubule extension at the cell posterior. To test this possibility, we performed immunofluorescence microscopy using the YL 1/2 antibody [29], which labels tyrosinated α-tubulins that are newly assembled into the microtubule corset [8]. The results showed that the elongated posterior of the CYC13-deficient and the CRK2-deficient cells was extensively stained by YL 1/2 (Fig 4A), indicating that the elongated posterior had increased amounts of tyrosinated microtubules. Notably, the elongated posterior of CYC13 RNAi cells and CRK2 RNAi cells often assumed a blunt end, as opposed to the tapered end of the posterior of the non-induced control cells (Fig 4A). Moreover, the multiple posterior branches of the CYC13-deficient and CRK2-deficient cells were also extensively stained by YL 1/2 (Fig 4A), demonstrating that the posterior branches all had increased amounts of tyrosinated microtubules. These observations suggest that posterior elongation was attributed to excessive extension of the corset microtubules.

The formation of multiple posterior branches in the CYC13-deficient and CRK2-deficient cells suggests that microtubule re-modeling might have occurred to organize a bundle of aligned plus-ends of the newly assembled microtubules at these posterior branches. To test this possibility, we tracked the distribution of the microtubule plus-ends-binding protein XMAP215, which is a well-characterized microtubule polymerase [30] and has been demonstrated to be a marker for microtubule assembly at the posterior cell tip in *T. brucei* [7].

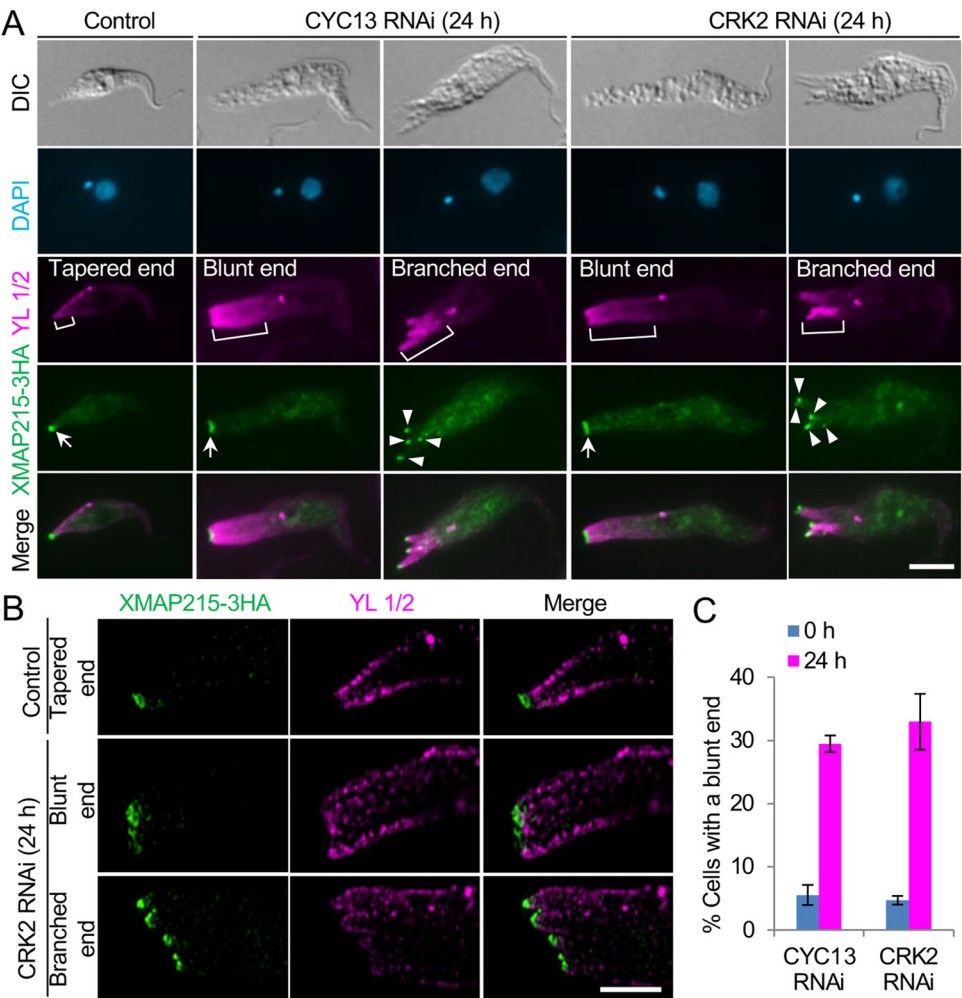

**Fig 4. CYC13 and CRK2 are required for maintaining posterior morphology.** (**A**). Immunofluorescence microscopy to examine the synthesis of new microtubules and the distribution of the microtubule plus-ends-binding protein XMAP215. Cells expressing endogenously 3HA-tagged XMAP215 were immunostained with FITC-conjugated anti-HA antibody and YL 1/2 antibody. The brackets outline the intensively stained posterior by YL 1/2 antibody, and the arrows indicate the XMAP215-enriched posterior tip and branched posterior tips. Scale bar: 5 μm. (**B**). 3D-SIM super-resolution microscopic analysis of the posterior tip of control and CRK2 RNAi cells. Cells expressing XMAP215-3HA were co-immunostained with FITC-conjugated anti-HA antibody and YL 1/2 antibody. Scale bar: 5 μm. (**C**). Quantitation of control, CYC13 RNAi, and CRK2 RNAi cells with a blunt posterior end. Error bars indicate S.D. from three independent biological replicates.

XMAP215 was epitope-tagged from one of its endogenous loci in both the CYC13 RNAi cell line and the CRK2 RNAi cell line, and immunofluorescence microscopy and 3D-SIM super-resolution microscopy showed that XMAP215 was detected at each of the multiple posterior branches (Fig 4A and 4B), demonstrating the formation of a bundle of microtubule plus-ends at each of the tips of the posterior branches. Notably, the XMAP215 fluorescence signal in the blunt end of the elongated posterior of the RNAi cells was much wider than that in the pointed end of the tapered posterior of the control cells (Fig 4A), and 3D-SIM showed that the blunt end of the elongated posterior appeared to possess multiple, discrete microtubule plus-ends (Fig 4B). Those 1N1K cells with a blunt posterior end containing multiple XMAP215 foci were increased from ~6% to ~30% after CYC13 RNAi for 24 h and from ~5% to ~33% after CRK2 RNAi for 24 h (Fig 4C). Taken together, these results suggest that when either CRK2 or

CYC13 was knocked down, the posterior end of the cells became elongated and often branched, which could be attributed to the deposition of tyrosinated (newly-synthesized) tubulins. Therefore, we postulate that the CRK2-CYC13 complex regulates posterior cytoskeleton morphogenesis by restricting excess microtubule extension and maintaining a tapered posterior end.

## Identification of β-tubulin as an *in vitro* substrate of CRK2

The drastic effect exerted by knockdown of CRK2 or CYC13 on the subpellicular microtubules at the posterior cell tip (Figs 1–4) prompted us to investigate the potential role of CRK2 in directly regulating the tubulin proteins, thereby affecting microtubule dynamics. A previous study reported that Cdk1, a mitotic cyclin-dependent kinase (CDK) in humans, phosphorylates β-tubulin on Ser-172, which inhibits its incorporation into microtubules [31]. We asked whether β-tubulin in *T. brucei* was a substrate of CRK2 and whether phosphorylation of β-tubulin by CRK2 might contribute to posterior cytoskeleton morphogenesis. We first performed GST pull-down to test whether CRK2 and β-tubulin interact, and the results showed that GST-fused β-tubulin was able to pull down CRK2 from trypanosome cell lysate (Fig 5A). We next performed *in vitro* kinase assay using recombinant GST-β-tubulin, GST-CRK2, and the kinase-dead mutant CRK2$^{K75R}$ purified from *E. coli*. The Lys-75 residue in CRK2 is an evolutionarily conserved residue in the kinase sub-domain II of all protein kinases, which is required for phosphotransfer reaction [32]. The results showed that CRK2, but not CRK2$^{K75R}$, was able to phosphorylate β-tubulin (Fig 5B), demonstrating that β-tubulin is an *in vitro* substrate of CRK2. It should be noted that CRK2 alone possesses kinase activity toward its substrate, but its binding to CYC13 enhances kinase activity [28]. Further, mass spectrometry analysis of GST-fused β-tubulin after *in vitro* kinase assay identified six phosphosites, Ser-18, Ser-115, Ser-172, Ser-248, Ser-351, and Thr-372 (Fig 5C, residues in blue, and S2 Fig). Ser-18 and Ser-115 were previously identified as *in vivo* phosphosites by a phosphoproteomics screen [33], which identified five additional *in vivo* phosphosites on β-tubulin (Fig 5C, residues in green). Among the *in vitro* CRK2 phosphosites and the *in vivo* phosphosites, only Ser-172 is followed by a proline residue (Fig 5D) and, thus, it meets the criteria of the consensus CDK phosphosite [34]. Further, three residues, Ser-172, Ser-351, and Thr-372, are evolutionarily conserved (Fig 5D), but phosphorylation of Ser-351 and Thr-372 has not been observed in any other organisms. Among the *in vivo* phosphosites on β-tubulin that are not *in vitro* CRK2 phosphosites, two residues, Ser-95 and Ser-285, are also evolutionarily conserved (Fig 5E), and the Ser-285 equivalent in human β-tubulin, Thr-285, is phosphorylated *in vivo* [35]. To further confirm the six CRK2 *in vitro* phosphosites identified by mass spectrometry, we mutated each of the six phosphosites to alanine and performed *in vitro* kinase assay. The results showed that mutation of each of these sites reduced the phosphorylation level by ~38–60% (Fig 5F), demonstrating that CRK2 phosphorylates β-tubulin on these six residues.

## Phosphorylation of β-tubulin on Ser-172 or Ser-351 impairs its incorporation into microtubules

Previously, it was reported that phosphorylation of β-tubulin on Ser-172 by Cdk1 in humans inhibits β-tubulin incorporation into microtubules [31]. The identification of multiple CRK2 phosphosites on β-tubulin (Fig 5), including the conserved Ser-172 residue, prompted us to investigate their roles in the incorporation of β-tubulin into microtubules in *T. brucei*. We focused on the phosphorylation of the three evolutionarily conserved residues, Ser-172, Ser-351, and Thr-372, and the two *in vivo* phosphosites, Ser-18 and Ser-115. However, when mutating the Thr-372 residue for functional analysis in *T. brucei*, we found that replacing Thr-

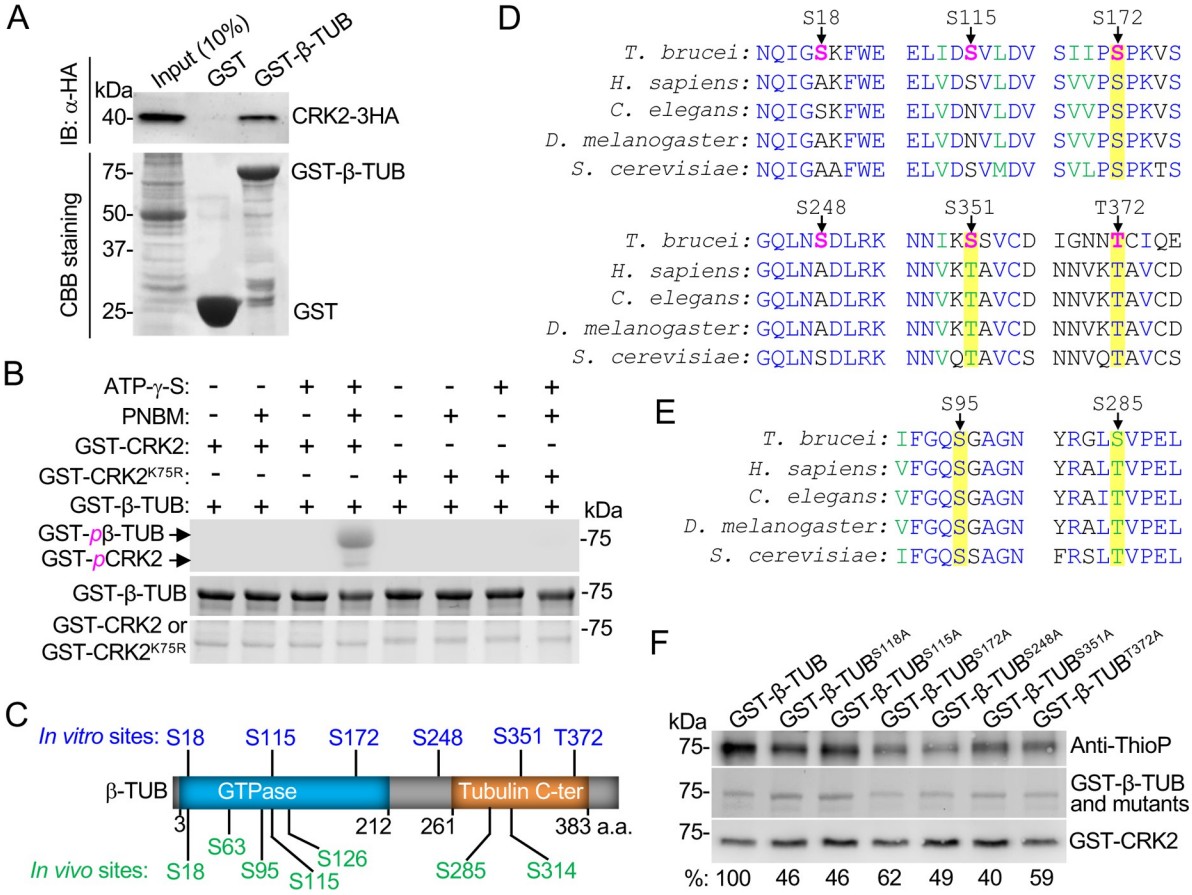

**Fig 5. Identification of β-tubulin as an *in vitro* substrate of CRK2.** (**A**). Interaction of β-tubulin with CRK2 examined by GST pull-down. CBB, Coomassie brilliant blue. (**B**). CRK2 phosphorylates β-tubulin *in vitro*. Shown is the *in vitro* kinase assay using purified recombinant GST-β-tubulin, GST-CRK2, and GST-CRK2$^{K75R}$. Phosphorylated GST-fused β-tubulin was detected by anti-ThioP antibody. GST-fused β-tubulin, CRK2, and CRK2$^{K75R}$ were stained by Coomassie brilliant blue. PNBM, p-Nitrobenzyl mesylate. GST-*p*β-TUB: phosphorylated GST-β-tubulin; GST-*p*CRK2: auto-phosphorylated GST-CRK2. (**C**). Schematic illustration of β-tubulin structural domains and the positions of *in vitro* CRK2 phosphosites (blue) identified by *in vitro* kinase assay and the *in vivo* phosphosites (green) identified by previous phosphoproteomics. (**D**). Sequence alignment of the six CRK2 phosphosites in *T. brucei* β-tubulin with that of human, *Drospophila*, *C. elegans*, and *S. cerevisiae* β-tubulin proteins. The evolutionarily conserved Ser-172, Ser-351, and Thr-372 residues are highlighted in yellow. (**E**). Sequence alignment of the two conserved *in vivo* phosphosites in *T. brucei* β-tubulin, Ser-95 and Ser-285, with that of human, *Drosophila*, *C. elegans*, and *S. cerevisiae* β-tubulin proteins. (**F**). Confirmation of the six *in vitro* CRK2 phosphosites by kinase assay. Recombinant GST-fused wild-type and mutant β-tubulin bearing the phospho-deficient (S/A or T/A) mutation for each of the six identified CRK2 phosphosites, which were stained with Coomassie brilliant blue, were used for kinase assay with purified GST-CRK2. The band intensity of the thio-phosphorylated wild-type and mutant β-tubulin proteins was presented as the percentage (%) relative to the thio-phosphorylated wild-type β-tubulin. The intensity of the thio-phosphorylated bands was normalized with that of Coomassie blue-stained protein bands. Shown are the averages of three independent biological replicates.

372 with several different residues all destabilized β-tubulin stability; therefore, this residue was not further characterized.

We first investigated the role of phosphorylation of Ser-172 by ectopically expressing the phospho-deficient (mutation to alanine) and phospho-mimic (mutation to aspartate) mutants of Ser-172 as well as the wild-type β-tubulin, which were each tagged with an N-terminal HA epitope, in the procyclic form of *T. brucei* using the tetracycline-inducible system [36]. Western blotting with anti-HA antibody confirmed the expression of HA-β-tubulin (hereafter referred to as "control tagged β-tubulin"), HA-β-tubulin$^{S172A}$, and HA-β-tubulin$^{S172D}$ at similar levels after tetracycline induction for 4 h (Fig 6A). However, after tetracycline induction for

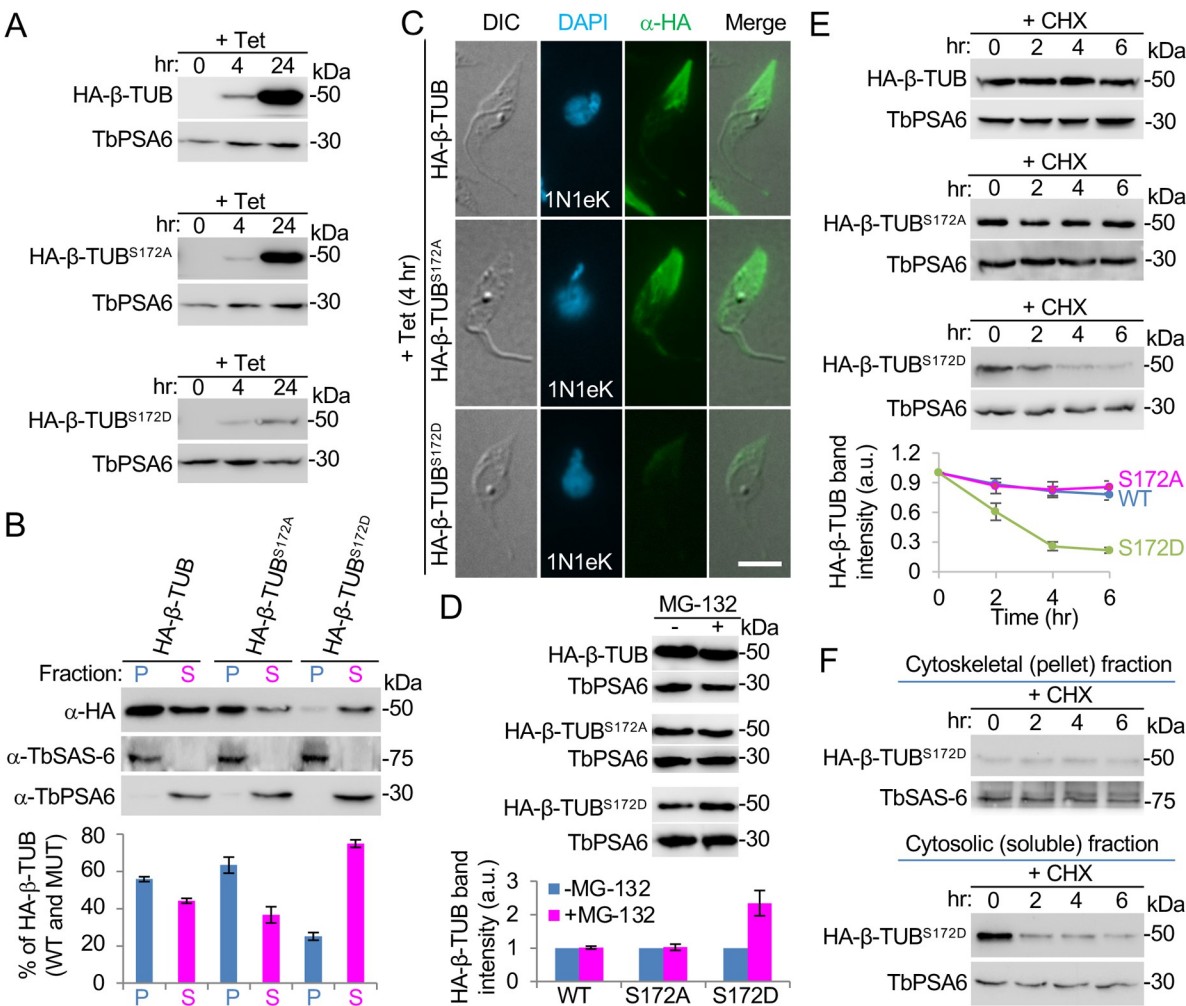

**Fig 6. Phosphorylation of β-tubulin on Ser-172 by CRK2 impairs its incorporation into cytoskeletal microtubules. (A)**. Western blotting to monitor the ectopically expressed control tagged β-tubulin, the S172A mutant, and the S172D mutant tagged with an N-terminal HA epitope. TbPSA6 served as a loading control. (**B**). Distribution of control tagged β-tubulin, the S172A mutant, and the S172D mutant in the cytosolic and cytoskeletal fractions of *T. brucei* cells. TbSAS-6 served as the cytoskeleton marker, and TbPSA6 served as the cytosol marker. The histogram shows the relative amounts of β-tubulin and its mutants, and error bars indicate S.D. from three independent biological replicates. (**C**). Incorporation of β-tubulin and its S172 mutants into the corset microtubules examined by immunofluorescence microscopy with FITC-conjugated anti-HA antibody. Scale bar: 5 μm. (**D**). Effect of MG-132 treatment on the stability of control tagged β-tubulin, the S172A mutant, and the S172D mutant. The histogram shows the band intensity of control tagged β-tubulin and β-tubulin mutants, which was normalized against the loading control TbPSA6. Error bars indicate S.D. (n = 3). (**E**). Stability of β-tubulin and its mutants examined by cycloheximide pulse-chase experiment. The histograms show the band intensity of β-tubulin and its mutants, which was normalized against the loading control TbPSA6 and then expressed relative to the first time point (0 hr). Error bars indicate S.D. from three independent biological replicates. (**F**). Stability of the S172D mutant in the cytosol and the cytoskeleton examined by cycloheximide pulse-chase experiment.

24 h, the level of the S172D mutant appeared to be much lower than that of the control tagged β-tubulin and the S172A mutant (Fig 6A). We first tested whether phosphorylation of Ser-172 affected the incorporation of β-tubulin into microtubules by analyzing the distribution in cytoskeletal and cytosolic fractions. Western blotting showed that ~56% of the control tagged β-tubulin was distributed in the cytoskeletal fraction (Fig 6B), whereas a slightly higher amount (~63%) of the S172A mutant and a much lower amount (~25%) of the S172D mutant were distributed in the cytoskeletal fraction (Fig 6B). It appeared that phosphorylation of Ser-172 impaired the incorporation of β-tubulin into the cytoskeletal microtubules and

dephosphorylation of Ser-172 promoted the incorporation of β-tubulin into the cytoskeletal microtubules. To further corroborate this finding, we performed immunofluorescence microscopy on detergent-extracted cytoskeletons of *T. brucei*. We induced the expression of the control tagged β-tubulin and mutant β-tubulins for 4 hours due to their similar expression levels (Fig 6A), and we focused on the cells at the S-phase of the cell cycle due to the active incorporation of β-tubulin into the growing cytoskeleton at the posterior portion of the cell. Immunofluorescence microscopic analysis of detergent-extracted cytoskeletons showed that the posterior portion of the cells expressing the control tagged β-tubulin or the S172A mutant was extensively immunostained, whereas the posterior portion of the cells expressing the S172D mutant was only lightly immunostained (Fig 6C), confirming that the S172D mutant was less efficiently incorporated into the cytoskeletal microtubules. After tetracycline induction for 8 h, the control tagged β-tubulin and the S172A mutant, but not the S172D mutant, were detectable throughout the cytoskeleton (S3 Fig), suggesting that HA-tagging of β-tubulin at its N-terminus does not affect its incorporation into microtubules. Taken together, these results suggest that phosphorylation of β-tubulin on Ser-172 inhibits the incorporation of β-tubulin into cytoskeletal microtubules.

The lower expression level of the S172D mutant than the control tagged β-tubulin and the S172A mutant after prolonged (24 h) tetracycline induction (Fig 6A) led us to hypothesize that the S172D mutant was being actively degraded in trypanosome cells. To test this hypothesis, we first treated the cells expressing the control tagged β-tubulin or the mutant β-tubulin with the proteasome inhibitor MG-132, and then examined the protein levels by western blotting (Fig 6D). The results showed that treatment with MG-132 exerted no effect on the levels of the control tagged β-tubulin and the S172A mutant, but it caused a ~2.3-fold increase of the level of the S172D mutant (Fig 6D), demonstrating that the steady-state levels of the S172D mutant are increased when proteasome activity is inhibited. To further confirm that the S172D mutant is being actively degraded, we carried out pulse-chase experiments to examine the degradation kinetics after inhibition of protein synthesis with cycloheximide. The results showed that the level of the control tagged β-tubulin was decreased by ~20% after 6 h, and the level of the S172A mutant was decreased by ~15% after 6 h of cycloheximide treatment (Fig 6E). However, the level of the S172D mutant was decreased by ~80% after cycloheximide treatment for 6 h (Fig 6E). The half-life of the S172D mutant was estimated to be ~2.6 h, whereas the half-life of the control tagged β-tubulin and the S172A mutant was much longer than 6 h (Fig 6E). These results demonstrated that the S172D mutation caused destabilization of β-tubulin. Further, we investigated the degradation kinetics of the S172D mutant in the cytosolic (soluble) and cytoskeletal (pellet) fractions, and the results showed that the S172D mutant in the cytosolic fraction, but not in the cytoskeletal fraction, was degraded after cycloheximide treatment (Fig 6F). Altogether, these results suggest that phosphorylation of β-tubulin on Ser-172 by CRK2 inhibits its incorporation into microtubules and subsequently destabilizes β-tubulin in the cytosol.

The corresponding position of the *T. brucei* β-tubulin Ser-351 residue in other organisms is occupied by a threonine residue (Fig 5D) and, thus, it is also considered to be conserved. To test the potential effect of Ser-351 phosphorylation on β-tubulin function, we ectopically expressed the S351A and S351D mutants of β-tubulin in *T. brucei* and examined their incorporation into the cytoskeletal microtubules and their stability. Western blotting showed that the level of the S351D mutant was much lower than that of the S351A mutant and the control tagged β-tubulin after tetracycline induction for 24 h (Fig 7A). Further, fractionation of the cytosolic and cytoskeletal fractions of cells showed that ~10% the S351D mutant was detected in the cytoskeletal fraction, compared to ~40% of the S351A mutant and ~55% of the control tagged β-tubulin detected in the cytoskeletal fraction (Fig 7B). Immunofluorescence microscopic imaging of detergent-extracted cytoskeletons showed that the fluorescence signal of the

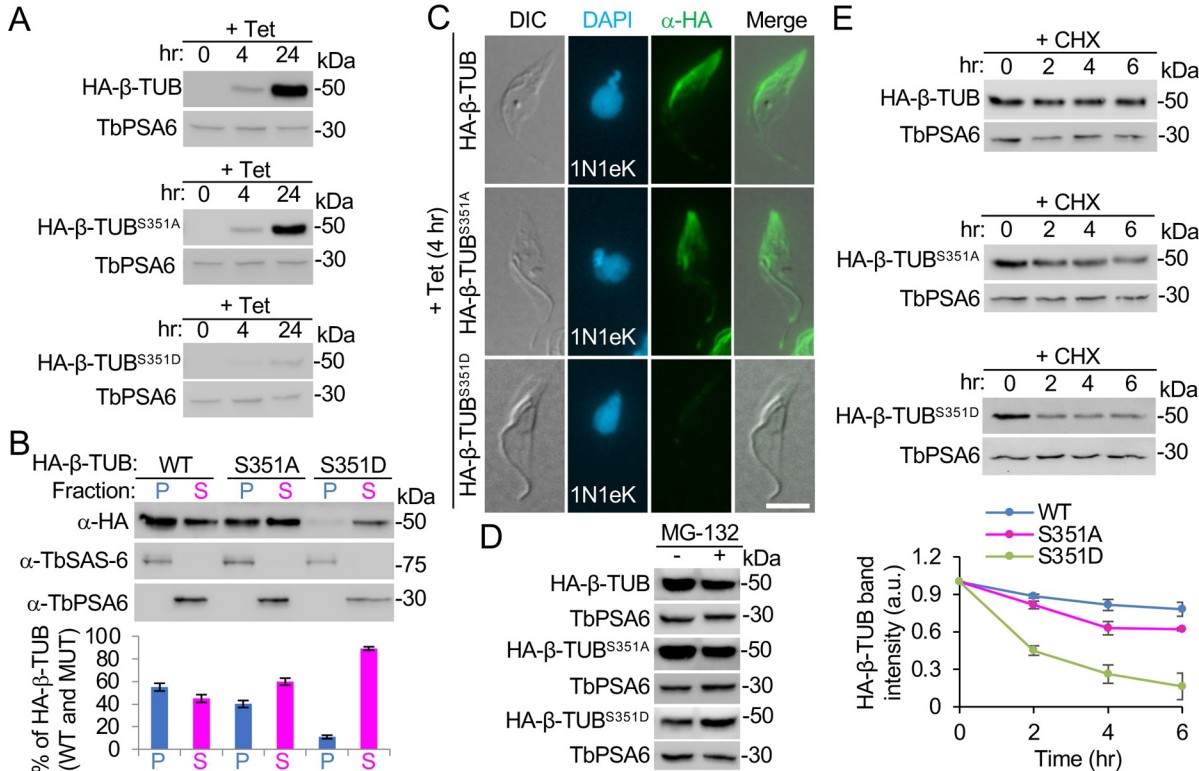

**Fig 7. Effects of Ser-351 phosphorylation on the incorporation of β-tubulin into microtubules.** (**A**). Ectopic expression of control tagged β-tubulin, the S351A mutant, and the S351D mutant in *T. brucei*. TbPSA6 served as a loading control. (**B**). Distribution of control tagged β-tubulin, the S351A mutant, and the S351D mutant in the cytosolic and cytoskeletal fractions of *T. brucei* cells. The histogram shows the relative amounts of control tagged β-tubulin and its mutants, and error bars indicate S.D. from three independent biological replicates. (**C**). Incorporation of control tagged β-tubulin, the S351A mutant, and the S351D mutant into the subpellicular microtubules examined by immunofluorescence microscopy with FITC-conjugated anti-HA antibody. Scale bar: 5 μm. (**D**). Effect of MG-132 treatment on the stability of control tagged β-tubulin, the S351A mutant, and the S351D mutant. TbPSA6 served as a loading control. (**E**). Stability of control tagged β-tubulin, the S351A mutant, and the S351D mutant examined by cycloheximide pulse-chase experiment. The histograms show the band intensity of the β-tubulin mutants, which was normalized against the loading control (TbPSA6) and then expressed relative to the first time point (0 hr). Error bars indicate S.D. from three independent biological replicates.

S351D mutant on the cytoskeleton was barely detectable and was much weaker than that of the S351A mutant and the control tagged β-tubulin (Fig 7C). These results demonstrated that S351D mutation inhibited the incorporation of β-tubulin into cytoskeletal microtubules. It should be noted that although the S351A mutant in the cytoskeletal fraction had lower amounts than the control HA-tagged β-tubulin (Fig 7B), its signal intensity on the cytoskeleton was not significantly different from that of the control tagged β-tubulin (Fig 7C). Therefore, the S351A mutation likely does not affect the incorporation of β-tubulin into the cytoskeletal microtubules. Further, treatment of cells with MG-132 caused an increased level of the S351D mutant (Fig 7D), and analysis of the degradation kinetics showed that the S351D mutant was rapidly degraded, with an estimated half-life of ~1.8 h (Fig 7E). It was also noted that the S351A mutant was degraded at a slower rate than the S351D mutant and a faster rate than the control tagged β-tubulin, albeit the S351A mutant and the control tagged β-tubulin both had an estimated half-life of >6 h (Fig 7E). Overall, these results suggest that phosphorylation of Ser-351 inhibits β-tubulin incorporation into cytoskeletal microtubules.

Finally, we investigated the potential role of two other CRK2 phosphosites, Ser-18 and Ser-115, because both sites were previously detected as *in vivo* phosphosites in *T. brucei* [33],

despite that they are not evolutionarily conserved (Fig 5D). Using immunofluorescence microscopy and western blotting, we found that both the phospho-deficient and the phospho-mimic mutants of Ser-18 and Ser-115 had comparable levels when ectopically expressed in *T. brucei* under a tetracycline-inducible promoter and were normally incorporated into cytoskeletal microtubules (S4 Fig), suggesting that their phosphorylation by CRK2 does not impact incorporation of β-tubulin into microtubules.

### Knockdown of CRK2 and CYC13 causes increased levels of cellular β-tubulin and α-tubulin

The finding that phosphorylation of β-tubulin on Ser-172 and Ser-351 by CRK2 targeted β-tubulin for degradation (Figs 6 and 7) led us to hypothesize that knockdown of CRK2 or CYC13 stabilizes β-tubulin and causes increased levels of cellular β-tubulin. To test this hypothesis, we endogenously tagged β-tubulin with an N-terminal triple HA epitope in the CRK2 RNAi cell line and the CYC13 RNAi cell line, and then performed immunofluorescence microscopy and western blotting. Immunofluorescence microscopy of detergent-extracted cytoskeletons showed that 3HA-tagged β-tubulin was distributed throughout the cytoskeleton in both the non-induced control and the RNAi-induced cells, despite the difference in cell morphology (Fig 8A). The 3HA-tagged β-tubulin was also detected in the flagellar axoneme and the spindle, and was fractionated into the cytoskeletal pellet only (S5 Fig). These results indicate that knockdown of CRK2 or CYC13 does not affect the incorporation of 3HA-β-tubulin into the cytoskeletal microtubules. Further, western blotting showed that the level of 3HA-β-tubulin was increased after RNAi of CYC13 and RNAi of CRK2 (Fig 8B). To corroborate

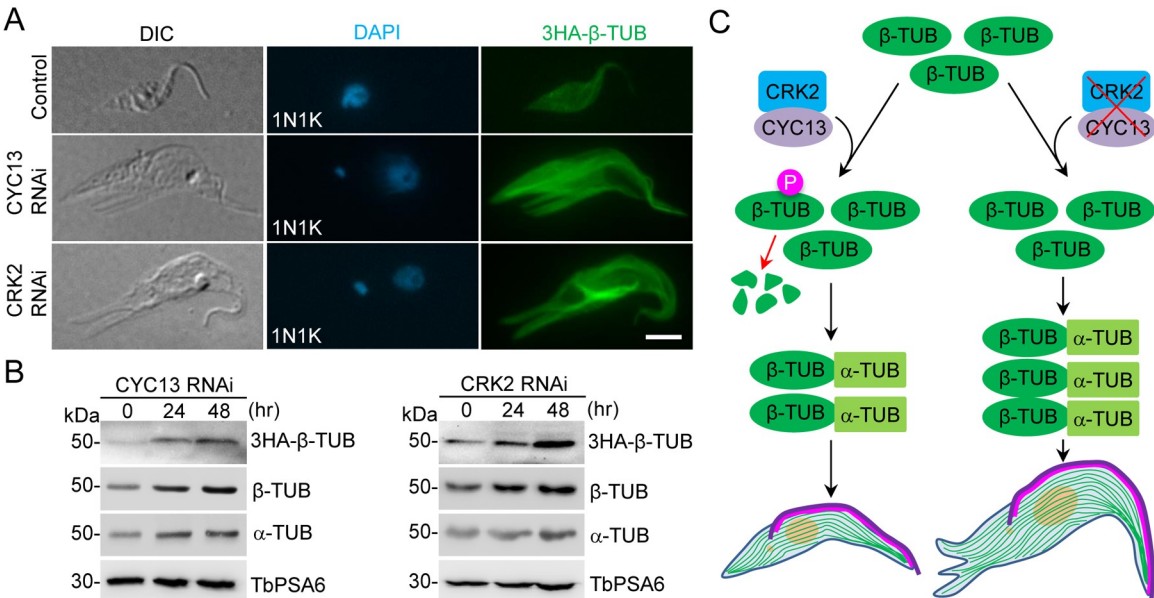

**Fig 8. Knockdown of CYC13 or CRK2 causes an increase of cellular β-tubulin levels.** (**A**). Immunofluorescence microscopic analysis of β-tubulin in the detergent-extracted cytoskeleton of non-induced control cells, CYC13 RNAi cells, and CRK2 RNAi cells. β-tubulin was endogenously tagged with an N-terminal triple HA epitope and detected by FITC-conjugated anti-HA antibody. Scale bar: 5 μm. (**B**). Western blotting to detect levels of 3HA-β-tubulin, β-tubulin, and α-tubulin in non-induced control cells, CYC13 RNAi cells, and CRK2 RNAi cells. An equal amount of total proteins for each time point was loaded into each well. 3HA-β-tubulin was detected by anti-HA antibody, β-tubulin was detected by KMX-1 antibody, and α-tubulin was detected by anti-α-tubulin antibody. TbPSA6 served as a loading control. (**C**). Model of CRK2-mediated phosphorylation of β-tubulin on its stability, microtubule dynamics, and cytoskeleton morphology in *T. brucei*. Green, subpellicular microtubule array; Pink, FAZ-associated microtubule quartet; Purple, flagellar axoneme.

this finding, we performed western blotting with the anti-β-tubulin antibody KMX-1 [37], and the results showed a similar increase of the level of native β-tubulin protein after RNAi of CYC13 or CRK2 (Fig 8B). We also examined the level of α-tubulin by immunoblotting with the anti-α-tubulin antibody, and the results showed that the level of α-tubulin was similarly increased after RNAi of CYC13 or CRK2 (Fig 8B). These results demonstrated that knockdown of the CRK2-CYC13 complex caused an accumulation of cellular β-tubulin and α-tubulin proteins.

## Discussion

We previously identified the CRK2-CYC13 complex as an essential S-phase CDK-cyclin complex to promote DNA replication by phosphorylating the CMG (Cdc45-Mcm2-7-GINS) complex subunit Mcm3 to facilitate CMG complex assembly in the procyclic form of *T. brucei* [28]. In this study, we extended the characterization of the CRK2-CYC13 complex and identified an additional function for this complex in regulating microtubule dynamics and cytoskeleton morphogenesis in *T. brucei*. This novel finding highlighted the essential involvement of CRK2 in diverse cellular processes during S-phase, such as DNA replication and posterior cytoskeleton morphogenesis. Therefore, trypanosomes need to coordinate these cellular processes during S-phase, and the identification of the CRK2-CYC13 complex in regulating both DNA replication and cytoskeleton morphogenesis in *T. brucei* suggests a key role of this CDK-cyclin complex in the coordinated control of multiple cellular events during S-phase progression.

Knockdown of CRK2 or CYC13 caused an excessive elongation and branching of the cell posterior (Figs 1–4). Posterior elongation was also observed in other cell cycle regulatory protein-deficient cells of the procyclic form, such as CYC2 RNAi cells [20,21], CYC7 RNAi cells [22], CRK1 RNAi cells [22], and CRK1-CRK2 double RNAi cells [23,24]. However, the cells with an elongated posterior in these previously reported RNAi cell lines were only about half of such cells after CRK2 RNAi or CYC13 RNAi reported in this study. Notably, RNAi of CRK2 alone in a previous study showed no growth defects and no posterior elongation phenotype [38], likely due to the low efficiency of RNAi in their study where a different CRK2 coding sequence and the double-stranded RNAi construct (pZJM) were used for RNAi. It is possible that the posterior elongation phenotype of the CRK1-CRK2 double RNAi cells reported previously [23] was attributed to the depletion of CRK1 only. Nonetheless, the CRK2 RNAi cell line generated in our lab by targeting a different CRK2 coding sequence and by using the stem-loop RNAi construct showed severe growth defects [28] and a significant accumulation of cells with an elongated and branched posterior after only 24 h (Fig 1B). In comparison, a significant accumulation of the cells with an elongated posterior was only observed after 4 days for CRK1 RNAi and CYC7 RNAi [22] and after 5 days for CRK1-CRK2 double RNAi [23]. Thus, our results suggest that among these cell cycle regulatory proteins, the CRK2-CYC13 complex plays a major role in regulating posterior cytoskeleton morphogenesis.

We provided evidence to demonstrate that posterior elongation and branching caused by RNAi of CRK2 or CYC13 was due to excessive microtubule extension and loss of microtubule convergence at the posterior cell tip (Figs 3 and 4). The extensive immunostaining of tyrosinated microtubules by YL 1/2 at the elongated posterior and the elongated and branched posterior of CRK2 RNAi cells and CYC13 RNAi cells (Fig 4) suggests altered microtubule dynamics. Similar phenotype was also discovered in the *T. brucei* tubulin polyglutamylase gene RNAi cells [18] and tubulin detyrosinase gene knockout cells [14], which provided additional evidence to support the notion that changes in microtubule dynamics at the cell posterior disrupt posterior cytoskeleton architecture in *T. brucei*. Thus, the CRK2-CYC13 complex

regulates microtubule dynamics at the cell posterior by restricting excessive microtubule extension and promoting microtubule convergence to facilitate the formation of a tapered posterior tip. In contrast, overexpression of TbZFP2 [19] or knockdown of TbSPHK [25] appears to only induce posterior elongation (the "nozzle" phenotype), without causing posterior branching. It suggests that these two proteins are involved in regulating microtubule dynamics, but not the organization of the microtubule plus-ends at the posterior cell tip. The mechanistic roles for these two proteins remain elusive, but they appear to be involved in the G1/S transition, similar to the G1 cyclins CYC2 [20,21] and CYC7 [22] and the G1 CDK homolog CRK1 [22,38]. It is possible that posterior elongation in these RNAi cells or TbZFP2-overexpression cells is attributed to G1 arrest, as previously proposed that posterior elongation is coupled to G1/S transition [23]. In this regard, these genes do not play a direct role in regulating microtubule dynamics, whereas CRK2-CYC13, tubulin polyglutamylases [18], and tubulin detyrosinases [14] directly regulate microtubule dynamics, as they modify tubulins by phosphorylation, polyglutamylation, or detyrosination.

CRK2 also appears to regulate the dynamics of the subpellicular microtubule array, the axonemal microtubules, and the FAZ-associated quartet microtubules. This is supported by the increase in cell size (or cytoskeleton size) and the increase in the length of the flagellum and its associated FAZ in CRK2 RNAi and CYC13 RNAi cells (S1 Fig). *T. brucei* duplicates its subpellicular microtubule array during the cell cycle in a semi-conservative manner [6], by which newly assembled microtubules are inserted into the existing microtubule array. In the absence of CRK2-mediated phosphorylation of β-tubulin, excess assembly of new microtubules occurred, and they were inserted throughout the subpellicular microtubule array, which may have caused the increase in the cytoskeleton size. The increase in the length of the flagellum and its associated FAZ is likely to be correlated with the increase in the overall cell length (Fig 2B, P-to-A length), driven by the elongation of the cell posterior (Fig 2B, K-to-P length), which appeared to have caused the migration of the basal body and its associated kinetoplast toward the cell posterior (Fig 2B, K-to-A length). The flagellar axonemal microtubules, and likely the FAZ-associated quartet microtubules as well, have an opposite polarity to those of the subpellicular microtubules [3]; therefore, the assembly/extension of these two microtubule structures occurs at the distal tips of the flagellum and the FAZ. Presumably, CRK2-mediated phosphorylation of β-tubulin could also affect the dynamics of the axonemal microtubules and the quartet microtubules. Thus, in CRK2-deficient and CYC13-deficient cells, the lack of β-tubulin phosphorylation may allow unrestricted extension of the axonemal microtubules and the quartet microtubules, leading to the elongation of the flagellum and the FAZ to coordinate with the increase in cell length driven by the excessive microtubule extension at the posterior cell tip.

The inhibitory effect of β-tubulin incorporation into microtubules by mutation of the CRK2 phosphosites Ser-172 or Ser-351 on β-tubulin to a phosphomimic residue (Figs 6 and 7) supports the essential role of CRK2 in regulating microtubule dynamics to inhibit microtubule extension. This notion is in agreement with the observed excessive microtubule extension at the posterior cell tip of the CRK2 RNAi cells and the CYC13 RNAi cells (Fig 4). In humans, Ser-172 of β-tubulin is phosphorylated by the mitotic cyclin-dependent kinase Cdk1, which impairs its incorporation into microtubules and is potentially involved in the regulation of microtubule dynamics during mitosis [31]. In neuron cells, Ser-172 of β-tubulin is phosphorylated by the MNB kinase, a dual-specificity tyrosine-regulated kinase (DYRK) in animals, and this phosphorylation regulates microtubule dynamics by inhibiting microtubule polymerization and is required for dendrite morphogenesis and neuron function [39]. Ser-172 is evolutionarily conserved (Fig 5D), and it resides in the T5 loop of β-tubulin near the ribose of the GTP nucleotide, where the presence of a bulky phosphate group might interfere with GTP binding and turnover, thereby inhibiting microtubule assembly [31]. Our results thus revealed

the conserved function of Ser-172 phosphorylation in regulating microtubule dynamics in *T. brucei*, although this phosphorylation is performed by an S-phase CDK, instead of a mitotic CDK. However, we cannot rule out the possibility that CRK3, the *T. brucei* mitotic CDK homolog, also phosphorylates β-tubulin on Ser-172 to regulate microtubule dynamics during mitosis. Further, we demonstrated that phosphorylation of β-tubulin on Ser-172 targeted β-tubulin for degradation in *T. brucei* (Fig 6), whereas it is unclear whether Cdk1 phosphorylation also destabilizes β-tubulin in humans. In this regard, phosphorylation of Ser-172 appears to serve as a signal to target β-tubulin for degradation, which may allow the cells to fine-tune the levels of β-tubulin to restrict excess microtubule extension, thereby preventing the elongation of the cell posterior (Figs 1–2), the increase of the cytoskeleton size, and the increase of the length of the flagellar axoneme and the microtubule quartet (S1 Fig). Phosphorylation of another evolutionarily conserved Ser-351 residue plays a similar role as phosphorylation of Ser-172 (Fig 7). Like the S172D mutant, the S351D mutant is also being actively degraded (Fig 7), suggesting that CRK2 phosphorylation of β-tubulin on either of the two sites is sufficient to target β-tubulin for degradation.

The identification of β-tubulin as an *in vitro* substrate of CRK2 (Fig 5) raised a question of where and when CRK2 phosphorylates β-tubulin in trypanosome cells. The CRK2-CYC13 complex functions as an S-phase CDK-cyclin complex [28], implying that CRK2 activity might peak during S-phase. The endogenous, epitope-tagged CRK2 and CYC13 appeared to be enriched at the posterior portion of the cell body during early cell cycle stages, with a pronounced concentration at the cell posterior end during S-phase, and somewhat were concentrated between the two segregated nuclei during post-mitotic phases (S6 Fig). It is possible that β-tubulin is phosphorylated by CRK2 at the posterior portion of the cell during the S phase of the cell cycle. Further, given that tubulins are incorporated into microtubules as α- and β-tubulin dimers, the observation that CRK2 phosphorylation of β-tubulin restricts excess microtubule extension and promotes β-tubulin degradation raised a question of whether CRK2 acts on β-tubulin alone or on both α- and β-tubulins and a question of whether phosphorylated β-tubulin is degraded alone or as a dimer of α- and β-tubulins. In cells depleted of CRK2 or CYC13, the levels of both α-tubulin and β-tubulin were increased (Fig 8B), suggesting that the stabilization of β-tubulin due to the lack of CRK2-mediated phosphorylation also caused the stabilization of α-tubulin, albeit the underlying mechanism is unknown. In one scenario, CRK2 might phosphorylate both α- and β-tubulins for their degradation to maintain optimal levels of both tubulins to prevent excess microtubule extension. Therefore, it would be interesting to test whether CRK2 also phosphorylates α-tubulin and whether this phosphorylation affects α-tubulin stability. In another scenario, CRK2 might only phosphorylate β-tubulin for degradation, whereby the turn-over of β-tubulin affects the fate of α-tubulin. The clustered organization of alternating α- and β-tubulin genes in the *T. brucei* genome [10,11] presumably ensures the expression of relatively equal amounts of both tubulin proteins, whereas in the absence of transcriptional gene regulation in *T. brucei*, how the fate of excess amounts of α- and/or β-tubulins is controlled remains unclear. Phosphorylation by CRK2 might act on the soluble pool of β-tubulin before they are incorporated into microtubules, thereby ensuring that there are no excess amounts of β-tubulin (and α-tubulin) proteins available in cells. Because α- and β-tubulins form dimers before being incorporated into microtubules, whether they regulate each other's stability according to certain circumstances is still unknown and requires further investigation.

The essential roles of CRK2 in promoting S-phase progression [28] and restricting excess microtubule extension raised a question of whether the two CRK2-regulated cellular processes are interdependent or whether the disruption of one process affects the other. Posterior cytoskeleton extension in *T. brucei* likely occurs during the early stages of the cell cycle (G1 to early

mitosis), but the most extensive growth of the posterior cytoskeleton appears to occur during G1 and S phases of the cell cycle [13]. Hence, the regulation of both processes by CRK2 suggests that the two processes are coordinated by CRK2, but the two processes are unlikely to be interdependent. It is possible that the prolonged S-phase progression in CRK2 RNAi and CYC13 RNAi cells may contribute indirectly to posterior elongation by giving cells more time to incorporate the excess tubulins into microtubules in the absence of CRK2-mediated phosphorylation. But if CRK2 does not act on β-tubulin directly to regulate microtubule dynamics, presumably the control mechanism for posterior cytoskeleton morphogenesis would still operate to restrict excess microtubule extension in the absence of CRK2, thereby generating S phase-arrested cells with normal cell posterior. Conversely, if CRK2 does not act on the DNA replication factor Mcm3 to promote DNA replication, presumably the regulatory machinery for DNA replication would still operate to promote S-phase progression in the absence of CRK2, despite that the cell posterior would still be excessively elongated. Therefore, CRK2 appears to coordinate the two cellular processes during S-phase by regulating cellular process-specific proteins.

In summary, we have discovered a control mechanism for cytoskeleton morphogenesis in *T. brucei* by which the S-phase CDK-cyclin complex CRK2-CYC13 phosphorylates β-tubulin to target it for degradation, thereby reducing the available pool of β-tubulin to restrict excess microtubule extension for maintaining cytoskeleton architecture (Fig 8C). In cells depleted of CRK2 or CYC13, theoretically none of the β-tubulin proteins are phosphorylated and, hence, none of the β-tubulin proteins are degraded (Fig 8C), leading to increased amounts of β-tubulin (Fig 8B). Additionally, the level of α-tubulin was proportionally increased in these RNAi cells (Fig 8B), presumably also due to stabilization, although the underlying mechanism is unknown. Consequently, microtubule extension is out of control in these RNAi cells, resulting in the elongation and branching of the posterior cytoskeleton, the increase in the cytoskeleton size, and the elongation of the flagellar axoneme and the FAZ-associated microtubule quartet (Fig 8C). Therefore, CRK2-mediated phosphorylation of β-tubulin fine-tunes the abundance of β-tubulin and α-tubulin proteins to prevent excess microtubule extension, thereby maintaining proper cytoskeleton morphology in *T. brucei*.

## Materials and methods

### Trypanosome cell culture and RNAi

The procyclic form of *T. brucei* 29–13 cell line [36] was cultured at 27˚C in the SDM-79 medium supplemented with 10% heat-inactivated fetal bovine serum (Atlanta Biologicals, Inc), 15 μg/ml G418, and 50 μg/ml hygromycin. Cells were sub-cultured by 1/10 dilution with fresh medium whenever the cell density reached 5×10⁶/ml. The CRK2 RNAi cell line and the CYC13 RNAi cell line have been previously reported [28], and the knockdown of CRK2 or CYC13 by RNAi was confirmed previously [28]; therefore these data were not presented in this work. RNAi was induced by incubating with 1.0 μg/ml tetracycline. Two independent clonal cell lines of each of the two RNAi cell lines were used for analysis, and we observed almost identical phenotypes. Only the results from the characterization of one clonal RNAi cell line were presented.

### Epitope tagging of proteins from the endogenous locus of the genes

Endogenous tagging of β-tubulin with an N-terminal triple HA epitope, CRK2 with a C-terminal triple HA epitope, CYC13 with a C-terminal PTP epitope, XMAP215 with a C-terminal triple HA epitope was carried out by using the PCR-based epitope tagging method [40]. Purified PCR products were used to electroporate the CRK2 RNAi cell line and the CYC13 RNAi cell

line, and transfectants were selected under 1 µg/ml puromycin. Correct tagging of the gene with the epitope was confirmed by PCR and subsequent sequencing of the PCR product. Transfectants were cloned by limiting dilution in a 96-well plate.

### Ectopic expression of HA-tagged wild-type and mutant β-tubulin proteins

To ectopically express wild-type and various mutant forms of β-tubulin in the procyclic form of *T. brucei*, we cloned β-tubulin gene into the pLew100 vector [36], which was modified to include an N-terminal HA epitope. The resulting plasmid, pLew100-HA-β-tubulin, was used for site-directed mutagenesis to mutate Ser-18, Ser-115, Ser-172, Ser-351, and Thr-372 to either alanine or aspartate, generating phospho-deficient or phospho-mimic mutants, respectively. All the resulting plasmids were sequenced to confirm the mutation. Expression of HA-tagged β-tubulin and its various mutants was induced with 1.0 µg/ml tetracycline for up to 24 hours, and cells were collected after tetracycline induction for 0 hour, 4 hours, 8 hours, or 24 hours for analysis.

For MG-132 treatment to analyze protein stability, expression of HA-tagged β-tubulin or its mutants was induced with 1.0 µg/ml tetracycline for 4 hours, and then cells were treated with 25 µg/ml MG-132 for an additional 4 hours. For cycloheximide treatment to analyze protein degradation kinetics, cells containing pLew100-HA-β-tubulin or mutants of β-tubulin were induced with 1.0 µg/ml tetracycline for 4 hours and then treated with 100 µg/ml cycloheximide for up to 6 hours.

### GST pull-down and western blotting

The full-length coding sequence of β-tubulin was cloned into the pGEX-4T-3 vector, and the resulting plasmid was used to transform the *E. coli* BL21 strain. Expression of recombinant proteins was induced with 0.2 µg/ml IPTG for 5 hours at room temperature (~22˚C). Cells were harvested and lysed by sonication in PBS containing 0.5% Triton X-100. Cell lysate containing recombinant GST-β-tubulin was incubated with 15 µl settled Glutathione Sepharose 4B beads (GE HealthCare), and bound proteins were washed three times with PBS containing 0.5% Triton X-100. CRK2-3HA was expressed in *T. brucei* from the endogenous locus, and cells expressing CRK2-3HA were suspended in 500 µl cell lysis buffer (25 mM Tris-HCl, pH 7.6, 150 mM NaCl, 1mM DTT, 1% Nonidet P-40, and protease inhibitor cocktail). Trypanosome cells were lysed by thorough sonication, and cell lysate was cleared by centrifugation. GST-β-tubulin bound to Glutathione Sepharose 4B beads was then incubated with cleared trypanosome cell lysate containing CRK2-3HA for 1 h at 4˚C with gentle rotation. Beads were washed six times with the cell lysis buffer (see above), and bound proteins were eluted by boiling for 5 minutes in 1× SDS sampling buffer. Eluted proteins were separated by SDS-PAGE, transferred onto a PVDF membrane, and immunoblotted with anti-HA monoclonal antibody (Sigma-Aldrich) to detect CRK2-3HA. GST protein was used as a control.

For other western blotting experiments, trypanosome cell lysate was separated by SDS-PAGE, and proteins were transferred onto a PVDF membrane and immunoblotted with either anti-HA monoclonal antibody (Sigma-Aldrich), KMX-1 (anti-β-tubulin) monoclonal antibody (EMD Millipore), anti-α-tubulin monoclonal antibody (Sigma-Aldrich), or anti-TbPSA6 (*T. brucei* proteasome subunit α6) polyclonal antibody [41].

### *In vitro* kinase assay using the thiophosphorylation method

Recombinant GST-β-tubulin, GST-fused β-tubulin mutants, GST-CRK2, and GST-CRK2$^{K75R}$ were expressed in *E. coli* and purified through a glutathione Sepharose 4B column (GE HealthCare). Recombinant proteins were eluted with 10 mM glutathione and dialyzed in PBS. *In*

*vitro* kinase assay was carried out with the method that uses a semisynthetic epitope to detect thiophosphorylated kinase substrates [42]. Purified recombinant GST-β-tubulin and its various mutants were incubated with GST-CRK2 or GST-CRK2$^{K75R}$ in the kinase assay buffer (10 mM HEPES pH7.5, 10 mM MgCl$_2$, 50 mM NaCl, and 2 mM DTT) containing 1 mM ATP-γ-S for 30 min at room temperature. Subsequently, 50 mM p-nitrobenzyl mesylate (PNBM), which alkylates thiophosphates to form thiophosphate ester epitopes that can be recognized by anti-thiophosphate ester antibody, was added to the kinase reaction and was incubated for 60 min at room temperature. Thiophosphorylated proteins separated and transferred onto a PVDF membrane were immunoblotted with the anti-thiophosphate ester (anti-ThioP) monoclonal antibody (1:5,000 dilution, ThermoFisher) [42].

### *In vitro* kinase assay and mass spectrometry

Purified recombinant GST-β-tubulin was mixed with purified recombinant GST-CRK2 in the kinase assay buffer (see above) containing 0.2 mM ATP for 30 min at room temperature. Kinase reaction was stopped by boiling the beads in 1× SDS sampling buffer for 5 min, loaded onto SDS-PAGE, and stained with Coomassie blue. To identify phospho-peptides, the gel slice containing GST-β-tubulin was excised. Excised protein band was digested with 160 ng trypsin for 4 h at 37°C, following published procedures [43], and peptides were extracted with 50 ml of 50% acetonitrile and 5% formic acid. Extracted peptides were dried using SpeedVac, resuspended in 2% acetonitrile and 0.1% formic acid, and injected onto Thermo LTQ Orbitrap XL (ThermoFisher Scientific), following published procedures [28]. Samples were analyzed on an LTQ Orbitrap XL interfaced with an Eksigent nano-LC 2D plus ChipLC system (Eksigent Technologies). Samples were loaded onto a ChromXP C18-CL trap column (200 mm i.d. x 0.5 mm length) at a flow rate of 3 nL/min. Reverse-phase C18 chromatographic separation of peptides was carried on a ChromXP C18-CL column (75 mm i.d x 10 cm length) at 300 nL/min. The LTQ Orbitrap was operated in a data-dependent mode to simultaneously measure full-scan MS spectra in the Orbitrap and the five most intense ions in the LTQ by CID, respectively. In each cycle, MS1 was acquired at a target value of 1E6 with a resolution of 100,000 (m/z 400) followed by top five MS2 scan at a target value of 3E4. The mass spectrometric setting was as follows: spray voltage was 1.6 KV, charge state screening and rejection of singly charged ion were enabled. Ion selection thresholds were 8,000 for MS2, 35% normalized collision energy, activation Q was 0.25, and dynamic exclusion was employed for 30s. Raw data files were processed and searched against the *T. brucei* proteome database using the Mascot and Sequest HT (version 13) search engines. The search conditions used were as follows: peptide tolerance of 10 p.p.m. and MS/MS tolerance of 0.8 Da, with two missed cleavages permitted and the enzyme set as trypsin.

### Immunofluorescence microscopy

Immunofluorescence microscopy was performed using our published procedures [28]. Briefly, *T. brucei* cells were settled on glass coverslips and fixed with cold methanol at -20°C. To prepare *T. brucei* cytoskeletons, cells settled on glass coverslips were treated with 1% Nonidet-P40 in the PEME buffer (100 mM PIPES-NaOH, pH6.9, 1 mM MgSO$_4$, 2 mM EGTA, 0.1 mM EDTA), and then fixed with 4% paraformaldehyde. Immunostaining was performed by incubating the fixed cells or the fixed cytoskeletons with the following primary antibodies: fluorescein isothiocyanate (FITC)-conjugated anti HA monoclonal antibody (clone HA-7, H7411, Sigma-Aldrich; 1:400 dilution), YL 1/2 monoclonal antibody [29] (1: 1,000 dilution), anti-Protein A polyclonal antibody (anti-ProtA, P3775, Sigma-Aldrich, 1:400 dilution), anti-CC2D polyclonal antibody (1: 2,000 dilution) [44], or the anti-PFR2 monoclonal antibody (clone

L8C4, 1:50 dilution) [45]. Subsequently, cells or cytoskeletons on the coverslip were washed with PBS, and then incubated with the following secondary antibodies: Cy3-conjugated anti-rat IgG, Cy3-conjugated anti-rabbit IgG, FITC-conjugated anti-rabbit IgG, or Cy3-conjugated anti-mouse IgG. Finally, cells or cytoskeletons were washed with PBS, mounted in the DAPI-containing VectaShield mounting medium (Vector Lab), and observed with an inverted fluorescence microscope (Olympus IX71).

## Three-dimensional structured illumination microscopy (3D-SIM) super-resolution microscopy

3D-SIM super-resolution microscopy was carried out according to our published procedures [46]. Briefly, cells expressing XMAP215-3HA were settled on the coverslips, treated with the PEME buffer containing 1% Nonidet P-40 to prepare cytoskeletons, which were then fixed in cold methanol (-20°C) and incubated in blocking buffer (1% BSA in PBS) at room temperature. Cytoskeletons were co-immunostained with FITC-conjugated anti-HA antibody (Sigma-Aldrich) and YL 1/2 antibody, followed by incubating with Cy3-conjugated anti-rat IgG. Cytoskeletons were imaged under the Nikon Super Resolution Microscope n-SIM E instrument (Nikon Instruments Inc., Americas) with a 100× lens equipped with 488 nm and 592 nm lasers. The acquired SIM images were analyzed by the NIS-Elements AR software.

## Scanning and transmission electron microscopy

Scanning electron microscopy was performed using the previously published procedures [47]. Briefly, cells were settled onto glass coverslips and fixed with glutaraldehyde. Cells were then dehydrated with alcohol and dried by critical point drying. Cells on the coverslips were coated with a 8-nm metal film (Pt:Pd 80:20, Ted Pella Inc) using a sputter coater (Cressington Sputter Coated 208 HR, Ted Pella Inc.). Cells were then examined using the Nova NanoSEM 230 (FEI) electron microscope.

Transmission electron microscopic analysis of whole-mount cytoskeletons of *T. brucei* cells was performed according to published procedures [6,48]. Non-induced control cells, CRK2 RNAi cells, and CYC13 RNAi cells were washed twice with PBS, settled onto grids, and treated with 1% Nonidet P-40 in the PEME buffer. Cells on the grids were then fixed with glutaraldehyde, stained with 1% uranyl acetate, and imaged using a JEOL 1400 transmission electron microscope.

## Data analysis

Measurement of the distance between the posterior cell tip, the anterior cell tip, the nucleus, and the kinetoplast was performed with the ImageJ software (http://imagej.nih.gov/ij/), and the data thus obtained was exported to the GraphPad Prism software for analysis. The error bars represent standard deviation (SD) from the mean of three independent biological replicates. The numerical data used in all Figs are included in S1 Data.

## Supporting information

**S1 Data. Excel spreadsheet containing, in separate sheets, the underlying numerical data for Fig panels 1B, 2B, 2D, 4C, 6B, 6D, 6E, 7B, 7E, S1B, and S1D.**
(XLSX)

**S1 Fig. Knockdown of CYC13 or CRK2 produces cells with increased cell size and elongated flagellum and FAZ.** (**A**). Microscopic analysis of a non-induced control cells, CYC13 RNAi cells, and CRK2 RNAi cells. Scale bar: 5 μm. (**B**). Measurement of the cell size of non-

induced control cells, CYC13 RNAi cells, and CRK2 RNAi cells. 100 1N1K cells for each time point were used for measurement. (**C**). Immunofluorescence microscopic analysis of the flagellum and its associated FAZ in non-induced control cells, CYC13 RNAi cells, and CRK2 RNAi cells. The flagellum was labeled with anti-PFR2 (clone L8C4) antibody, and the FAZ was labeled with anti-CC2D antibody. Scale bar: 5 μm. (**D**). Measurement of the length of the flagellum and the length of the FAZ in non-induced control cells, CYC13 RNAi cells, and CRK2 RNAi cells. RNAi was induced for 48 h.
(PDF)

**S2 Fig. Mass spectrometry data of the CRK2 phosphosites on β-tubulin.**
(PDF)

**S3 Fig. Incorporation of ectopically expressed HA-β-tubulin into microtubules after longer times of tetracycline induction.** Shown are immunofluorescence microscopy images of HA-tagged β-tubulin and the S172A and S172D mutants after tetracycline induction for 8 hours.
(PDF)

**S4 Fig. Phosphorylation of β-tubulin on Ser-18 and Ser-115 exerts no effect on β-tubulin incorporation into cytoskeletal microtubules.** (**A**). Western blotting to detect the ectopically expressed control tagged β-tubulin, the S18A mutant, the S18D mutant, the S115A mutant, and the S115D mutant tagged with an N-terminal HA epitope. TbPSA6 served as a loading control. (**B**). Incorporation of control tagged β-tubulin and its mutants into the corset microtubules examined by immunofluorescence microscopy. Detergent-extracted cytoskeletons of *T. brucei* cells expressing HA-tagged β-tubulin or its mutants were immunostained with the FITC-conjugated anti-HA antibody. Scale bar: 5 μm.
(PDF)

**S5 Fig. Endogenously epitope-tagged β-tubulin is incorporated into microtubules of cytoskeleton, flagellar axoneme, and spindle.** (**A**). Immunofluorescence microscopic analysis of 3HA-β-tubulin in intact cells and detergent-extracted cytoskeletons. A, axoneme; S, spindle. Scale bar: 5 μm. (**B**). Distribution of endogenous 3HA-β-tubulin in the cytosolic and cytoskeletal fractions of *T. brucei* cells. TbPSA6 served as the cytosol marker. α-tubulin served as the cytoskeleton marker.
(PDF)

**S6 Fig. Subcellular localizations of endogenously epitope-tagged CRK2 and CYC13 during the cell cycle.** CRK2 was endogenously tagged with a triple HA epitope and CYC13 was endogenously tagged with a C-terminal PTP epitope. Cells were immunostained with the FITC-conjugated anti-HA monoclonal antibody and anti-Protein A polyclonal antibody, and counterstained with DAPI. Scale bar: 5 μm.
(PDF)

## Acknowledgments

The content is solely the responsibility of the authors and does not necessarily represent the official views of the National Institutes of Health. We thank Dr. Cynthia Y. He of National University of Singapore for providing anti-CC2D antibody, Dr. Keith Gull of University of Oxford for providing anti-PFR2 antibody (clone L8C4), Dr. Li Li of the Clinical and Translational Proteomics Service Center at University of Texas Health Science Center at Houston for assistance with mass spectrometry, Dr. James Gu of Houston Methodist Research Institute for

assistance with scanning electron microscopy, Dr. Heidi Kaplan of the University of Texas Health Science Center at Houston for assistance with transmission electron microscopy, and Dr. Kieu T.M. Pham of University of Texas Health Science Center at Houston for assistance with 3D-SIM super-resolution microscopy.

## Author Contributions

**Conceptualization:** Kyu Joon Lee, Ziyin Li.

**Formal analysis:** Kyu Joon Lee.

**Funding acquisition:** Ziyin Li.

**Investigation:** Kyu Joon Lee, Qing Zhou.

**Methodology:** Kyu Joon Lee, Qing Zhou.

**Project administration:** Ziyin Li.

**Supervision:** Ziyin Li.

**Validation:** Kyu Joon Lee.

**Visualization:** Kyu Joon Lee, Qing Zhou.

**Writing – original draft:** Ziyin Li.

**Writing – review & editing:** Kyu Joon Lee, Ziyin Li.

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
