## [Decision Letter · Decision Letter 0]

19 Dec 2022

Dear Dr. Li,

Thank you very much for submitting your manuscript "CRK2 controls cytoskeleton morphogenesis in Trypanosoma brucei by phosphorylating β-tubulin to regulate microtubule dynamics" for consideration at PLOS Pathogens. As with all papers reviewed by the journal, your manuscript was reviewed by members of the editorial board and by several independent reviewers. In light of the reviews (below this email), we would like to invite the resubmission of a significantly-revised version that takes into account the reviewers' comments.

We cannot make any decision about publication until we have seen the revised manuscript and your response to the reviewers' comments. Your revised manuscript is also likely to be sent to reviewers for further evaluation.

Sincerely,

Keith

Keith R. Matthews

Guest Editor

PLOS Pathogens

Margaret Phillips

Section Editor

PLOS Pathogens

Kasturi Haldar

Editor-in-Chief

PLOS Pathogens

orcid.org/0000-0001-5065-158X

Michael Malim

Editor-in-Chief

PLOS Pathogens

orcid.org/0000-0002-7699-2064

Reviewer's Responses to Questions

**Part I - Summary**

Reviewer #1: This manuscript is well written and there are some nice experiments testing the importance of CRK2 CYC13 in regulation of B-tubulin into the cytoskeleton. The first three results figures are very reasonable and the authors do show that knockdown of CYC13 and CRK2 by RNAi results in an increase of longer posterior ends, blunt ends and accumulation of tyrosinated tubulin indicative of new microtubules. The organisation of the extreme posterior end looks abnormal, but, surprisingly, XMAP215 is still localised into discrete foci, suggesting quite a level of microtubule minus end organisation even if there are multiple ends. The conclusion that CRK2-CYC13 might be regulating microtubule growth at the posterior end is reasonable, but regulation is quite hard to really test. Next, the authors tested if B-tubulin was a substrate of CRK2 and reported on six residues that reduced the level of phosphorylation. Two residues were extensively tested and found to result in a decrease in the ability of B-tubulin to incorporate into microtubules and the authors conclude that there is an increase in cellular levels of B-tubulin in the CRK2 and CYC13 RNAi cells. However, since the cells are larger overall, this might not be a correct conclusion. Whilst this is a nice piece of work extending their initial characterisation, it does lack impact for the target audience for PLOS pathogens.

Reviewer #2: In this manuscript, Lee et al. revisit the phenotype of two knockdown trypanosome cell lines, namely of the kinases CRK2 and CYC13. Through careful investigation combining light and electron microscopy, they show that the posterior end of these cells elongates substantially, with a clear extension of microtubules, often resulting in spectacular posterior rend branching. In the second part of the manuscript, they search for a mechanistic explanation to this phenomenon. As a working hypothesis, they rely on the work of Didier Job’s group who showed that beta-tubulin phosphorylation by CDK1 inhibits its incorporation in microtubules (ref 22). They produce a set of convincing data that beta-tubulin can be a substrate of CRK2 in vitro and then proceed to analyse the impact of several phosphorylation sites in trypanosomes, selected by a combination of criteria (experiments presented here, phospho-proteomic data, evolutionary conservation). Phosphorylation of two of these residues turned out to increase tubulin turnover and the authors propose a nice model whereby absence of CRK2 would slow down beta-tubulin turnover, allowing further extension of microtubules, hence providing a molecular explanation to the excessively long posterior end of knockdown cells. The work is of high quality and should appeal to parasitologists but also cell biologists interested in cytoskeleton formation. However, it faces a technological challenge that is discussed below, with possible suggestions. There are also some major implications of this work that would deserve to be discussed in the manuscript.

Reviewer #3: There have recently been a number of papers documenting morphological phenotypes of the trypanosome cell posterior. While a range of different proteins have been implicated in generating this phenotype, the control mechanisms are unknown. In this study, Lee et al., make an intriguing mechanistic connection between this phenotype and the action of a mitotic kinase complex, which acts to regulate the incorporation of tubulin at the cell posterior.

The work builds on earlier studies which noted that depletion of the mitotic kinase CRK2 has an effect on the cell posterior end. The Li group recently demonstrated that CRK2 interacts with CYC13, and in this study they revisit the posterior end phenotype in light of this interaction. They show that depletion of either CRK2 or CYC13 produces the same phenotype, and elegantly show that the cell size changes can be attributed solely to an elongation of the cell posterior. A combination of fluorescence microscopy and electron microscopy is used to show that this posterior end elongation can be traced to increased deposition of newly-synthesised (i.e. still tyrosinated) tubulin, and that the microtubule plus ends are now extended in parallel instead of being gathered together to generate the normal tapered cell posterior.

They subsequently show that beta-tubulin can pull down CRK2 from cell lysates, and that CRK2 can phosphorylate specific sites on beta-tubulin in vitro. Mutagenesis of two of these sites to mimic phosphorylation results in lower levels of expressed protein and reduced incorporation into the cytoskeleton - effects that they show are likely due to a shorter half-life brought on by increased proteasomal degradation of the phosphomimic proteins. Lastly, they obtain evidence that when CRK2 or CYC13 are depleted, the cellular levels of beta-tubulin are increased.

This is a very attractive mechanism, and one that has been previously documented in mammalian cells using a similar set of approaches (Fourest-Lieuvin et al., 2006). What is intriguing is that the same mechanism is being used here for maintenance of cell morphology, and is apparently being coordinated by CRK2 instead of CDK1.

It is a nice, self-contained story that suggests lots of interesting follow-up questions. The data are of good quality throughout, rigorously quantified, and feature an attractive blend of techniques. It will undoubtedly be of great interest to the trypanosomatid community, and a valuable addition to the growing body of work on the mechanisms governing cytoplasmic microtubule organisation and cell posterior end morphology.

I have a couple of modifications which I think are essential, as well as a longer list of more minor changes and editorial alterations that the authors may wish to take on board, and which I think would strengthen the manuscript.

Page numbers refer to those of the 26-page manuscript rather than the 38-page PDF.

**Part II – Major Issues: Key Experiments Required for Acceptance**

Reviewer #1: My main concern there is no evidence to show that cells are arrested in S-phase as outlined at the start of the results section and this is very important to show.

The timings chosen for analysis are a long time after initiation of RNAi and the phenotype of large posterior ends could be an accumulation over time leading the phenotypes observed rather than problems with the regulation of B-tubulin incorporation.

Reviewer #2: The authors used an ectopic copy of N-terminally HA-tagged beta tubulin under the control of a tetracycline operator to express control or point mutations of the candidate phosphorylated residues (Figures 5 and 6). They call the HA-tagged tubulin “wild type” tubulin. However, this is somehow misleading because this protein seems to behave not exactly as wild-type. The fractionation experiments indicate that ~40% of HA-beta-tubulin is retained in the soluble pool, in contrast to wild-type beta-tubulin for which the large majority is incorporated into microtubules and the soluble pool is very low (see for example ref. 10). This is also an issue for Figure 7 where the authors used the same HA-tagged tubulin version to evaluate beta-tubulin abundance and to correlate it with microtubule extension. If close to half of the tagged protein were in the soluble pool as shown in control conditions, this weakens the argument. We suggest probing directly for endogenous beta-tubulin using the commercially available beta-tubulin specific antibody KMX-1 (Birkett et al 1985). Looking for alpha-tubulin would also be relevant (see below), for example with the TAT-1 antibody (Woods et al JCS89).

IFA images reveal that after 4 hours of expression, this HA-tagged beta tubulin is incorporated at the posterior part of the cytoskeleton but with a pronounced lateral bias visible on all images, including for the version with mutated residues. This might reflect specific aspects of beta-tubulin incorporation but this would be surprising since the profile looks different from tyrosinated alpha-tubulin that is being used as a classic marker of microtubule assembly (ref. 4). This is a matter of concern for the interpretation of the data as it questions whether the tagged version is indeed representative of the wild-type beta-tubulin.

Having said that, about 60% incorporation in microtubules is so far the best result for a tagged tubulin in trypanosomes. Tagging alpha-tubulin turned out to be even more challenging with most of the tagged protein remaining in the soluble fraction (Bastin et al 96, Sheriff et al 2014).

We think that calling this tagged beta-tubulin “control tagged tubulin” (or something equivalent) would be more appropriate. We suggest showing expression after longer induction periods (24 hours or more). If it leads to a homogenous staining of the cytoskeleton, this would be very reinsuring.

Second, there is no evidence that this is in fact an overexpression. There are roughly 40 copies of beta-tubulin genes in the genome and from the presented data we cannot infer if induced expression of the HA-tagged protein(s) is leading to a notable increase in overall beta-tubulin. We can suggest a control experiment with the commercially available beta-tubulin specific antibody KMX-1 (Birkett et al 1985) that works very well in trypanosomes. The presence of the tag should allow separation of the HA-protein from the endogenous beta-tubulin and will reveal the ratio between them on western blots.

Similarly, western blot analysis with antibodies recognizing PTM’s on beta-tubulin (glutamylation or detyrosination, see Casanova IJP15, van der laan et al. Cell reports 2009) would show that the tagged protein behaves reasonably normally.

Finally, we could not find a description on which tetracycline inducible system the authors used in this study. Please provide details.

Points for discussion.

The discussion section is a bit repetitive with the results, it could refocused to include three points that sound quite exciting to us.

The first is about alpha-tubulin. Beta-tubulin is not incorporated alone but always as a dimer with alpha-tubulin. This study focuses solemnly on beta-tubulin as the controller of integration. It is known that soluble fraction of both alpha and beta tubulin is low (ref 10). This begs the question how overabundance of beta-tubulin alone can lead to such a striking phenotype while the level of alpha-tubulin is presumably unchanged? Is phosphorylated tubulin degraded as a dimer or just beta-tubulin alone? This has implications on the abundance of tubulin-dimers during RNAi. Since microtubules elongate excessively in the knockdown experiments, these results suggest that either (1) it is the turn-over of beta-tubulin that decides that of alpha tubulin; (2) that both alpha- and beta- would be targets of CRK2 or (3) that alpha- and beta-tubulin regulate each other according to circumstances as they live as dimers. We are not asking for experimental evidence here as this would take too much time, but this point should be discussed.

Second thing, what controls the timing of expression/activity of CRK2? We imagine it is cell cycle-regulated, is there some evidence, maybe from previous work of the authors, that it peaks towards the end of the cell cycle? We know it is difficult to synchronise trypanosome cultures, but some localisation data might be available. If this is not the case, it is fine to elaborate in the discussion.

Third, in CRK2/CYC13 RNAi conditions, it is assumed that tubulin production is constant while integration is higher. The extension of microtubules however, would still be limited by the production of its main building blocks. Since soluble pool of tubulin is low, in order to facilitate the observed posterior elongation, the cell cycle length would probably need to be longer than usual. Unless lack of phosphorylation-mediated-degradation a) acts on the dimer and b) is substantial enough to deliver sufficient material to facilitate the observed elongation. Cell cycle disruption is observed when we consult the characterization of this RNAi cell line (ref. 15) where most cells stay in the 1K1N stage (CRK2 KD) and the growth is slowed down after the addition of tetracycline. One valid hypothesis is here presented: Decrease in phosphorylation leads to increase in tubulin-integration (and decrease of degradation) and therefore posterior elongation as well as cells failing to assemble a tapering end. However, it would be interesting to read a discussion on how a prolonged S-phase could contribute to posterior microtubule elongation. Is the cell cycle disrupted because of posterior elongation as a consequence of increased incorporation (and decreased degradation) or does the disrupted cell cycle facilitate excessive posterior elongation and branching by giving the cells more time to produce excess tubulin during S-phase and integrate it in the absence of phosphorylation?

Reviewer #3: 1. P5 L12 I think an extra paragraph needs inserting here to provide more background context for the reader and to more fully acknowledge previous work in this area. The effect of cyclin depletion on the cell posterior was first noted in work done by the Mottram group (Hammarton et al., 2004 doi: 10.1074/jbc.M401276200), and the CRK2 phenotype was first observed by the Wang group (Tu & Wang, 2005; Tu et al., 2005 doi: 10.1091/mbc.e04-05-0368, 10.1242/jcs.02567). The de Graffenried group has also published two recent papers characterising proteins operating to maintain the morphology of the cell posterior (Hilton et al., 2018; Sinclair et al., 2021 - doi 10.1111/mmi.13986 and 10.1371/journal.ppat.1009588), and the Ersfeld group has (as already noted) characterised the effects of perturbing tubulin polyglutamylation - all this is essential background context for the reader and needs to come in the Introduction and not later on. P5 L24-26 (which are actually providing introductory information rather than results) can be absorbed into this paragraph. P13 L13-17 should also be relocated primarily to the Introduction. The original and interesting contribution of the work here is that it revisits the CRK2 phenotype in light of the more recent work from the Ersfeld and de Graffenried groups, and following the Li group's recent demonstration that CRK2 interacts with CYC13. It is of interest because it directly links the maintenance of the microtubule corset with cell cycle signalling complexes.

2. P9 L22+ I think it is essential that the authors here acknowledge (as, in fairness, they later do in the results and discussion) that phosphorylation of S172 in by a mitotic kinase was previously shown to affect the incorporation of beta-tubulin into microtubules in mammalian cells (Fourest-Lieuvin et al., 2006 - ref. 22). This is a plausible and attractive hypothesis to pursue, and it means that in their subsequent experiments they are demonstrating that a previously-observed mechanism also is operating in trypanosomes. What is interesting about their data is that it appears to be CRK2 and not CDK1 that is responsible for the phosphorylation event, and this is something they should comment on explicitly in their Discussion.

**Part III – Minor Issues: Editorial and Data Presentation Modifications**

Reviewer #1: Mainly with the first figure, but generally clearly written.

Page 6 line 3: It is not clear how the authors determined that the induced RNAi cells were arrested at S-phase and I could not find any supporting data.

Page 6 line 3-6: the description does not match figure 1B which is reported elongated or branched, but wording is describing posterior elongation.

Page 6 line 7-8 you should add the N number of the cells measured.

Page 6 Fig 1 measurements. This needs a cartoon image for readers to understand. Your measurements show that K to A is also significantly longer, so does that suggest that the anterior end is also longer?

Reviewer #2: The end of the introduction (p5 line 12) is very abrupt, we suggest to introduce a paragraph to describe the previous work on CRK2/CYC13 and to explain the goals of the study. It will also fluidify the beginning of the result section (bottom page 5).

Page 5, lane 4. For non-trypanosome people, please mention all copies of tubulins are identical (no sequence variants as observed in Plasmodium or Drosophila for example).

Page 5, line 9. Please note that the impact of tubulin detyrosination in trypanosomes has been investigated recently (van der laan et al. Cell reports 19).

Page 13, line 15. These experiments were done by RNAi gene knockdown, not by gene depletion (same comment page 14, line 15).

Figure 1:

A cartoon showing the change of distance between cellular landmarks in uninduced vs. induced conditions could help the reader to interpret the data depicted a bit quicker.

We apologise for having written such a long review, but the paper is very interesting and we hope our suggestions will be helpful.

Daniel Abbühl and Philippe Bastin

Reviewer #3: Minor points: (in manuscript order)

P4 L4. I would mention also that different tubulin isoforms, as well as PTMs, contribute to functional diversity. It's ok to focus on PTMs from then on.

P4 L15 I would recommend citing Sinclair & de Graffenried 2021 (doi: 10.1016/j.pt.2019.07.008) either additionally to or instead of the 1999 Gull one - the 2021 review is more up-to-date and focuses specifically on the corset.

P4 L17 I would recommend citing Robinson & Gull, 1995 (doi: 10.1083/jcb.128.6.1163) here instead of the review article.

P4 L19 I would recommend additionally citing Wheeler et al., 2013 (doi:10.1111/mmi.12436) here.

P5 L2 Just as a comment - as far as I'm aware, the polarity of the quartet microtubules has not been experimentally demonstrated, but they are assumed to have their plus ends at the cell anterior tip.

P5 L7 It might be worth commenting on how similar the clusters of alpha and beta tubulin genes are, if this information is readily available. This could have important consequences for post-translational modification, especially if the phosphorylation sites are not universally conserved.

P5 L13 Maybe the authors should specify explicitly that they recently identified CYC13 as being in a complex with CRK2? This provides a rationale for investigating both, and revisiting the depletion phenotype.

P5 L27. This statement needs clarifying. They authors show that CYC13 depletion results in the generation of cells with elongated or branched posteriors. This is a new observation. The morphological phenotype is similar to that documented following CRK2 depletion, which was already known. They then show that they can recapitulate previous descriptions of this phenotype in their RNAi cells.

P6 L3. The authors could maybe explicitly define what they mean by "elongation" (twofold increase in cell length?). This is important because there are no negative controls in Fig 1B, and they therefore need to explain why these are presumably scoring 0% according to their classification.

P6 The analysis presented in Fig 1CDE is impressive and very helpful.

P7 L23 I would recommend also citing Kilmartin et al., 1982 here (characterisation of YL1/2).

P7 L24. I think this could be clarified a bit. It's probably more accurate to say that the images show that the RNAi cells have increased amounts of tyrosinated tubulin at the elongated posterior ends. The data in Fig2 suggest (I think) that the number of microtubules present is roughly the same as in controls; "filled with" could be read as meaning that the number of microtubules increases.

P8 L18. I think this statement could perhaps be made more precisely. What the authors have shown is that when either CRK2 or CYC13 are depleted, the posterior end of the cell becomes elongated and that this can be attributed to the deposition of tyrosinated (newly-synthesised) tubulin.

P9 L1-2 and P24 L1. I wouldn't call this an "in vitro" interaction as the input is cell lysate rather than purified tubulin. It would be more accurate to say that beta-tubulin can pull down CRK2 from cell lysates. Was any attempt made to do a reciprocal pulldown, or demonstrate the interaction using a different technique?

P9 L3. A little bit more detail on the mutagenesis would be helpful. Has this mutant been used before and what is the role of the residue that is mutated? It would also be useful if a bit more detail on the kinase assay was provided - how is the phosphorylated product detected?

P10 L24 Maybe say "being actively degraded" instead of "an unstable protein"? That's probably more precise. Similarly in P11 L2 it might be better to say that steady-state levels of the mutant protein are increased when proteasome activity is inhibited.

P12 L22-23 Given that the immunofluorescence microscopy is being done on whole cells, and given too that Figures 5B, 6B showed a substantial cytoplasmic fraction of beta-tubulin, I don't think Figure 7A provides information on whether incorporation of beta-tubulin into the corset microtubules is being affected. The authors would need to examine extracted cytoskeleton preparations to be more sure of that.

P13 L16 The authors may wish to reconsider linking the CCCH zinc finger phenotype with the one described here. While clearly related to the CRK2 phenotype and worth mentioning in that context, the posterior "nozzle" phenotype caused by overexpression of the CCCH zinc finger protein does not involve any disruption of the organisation/gathering of the plus tips of the microtubules and so branching or extreme dilation of the posterior end is not seen (as far as I'm aware). This suggests that the phenotype seen with CRK2 (and other proteins causing branching/dilation of the cell posterior) results not just from elongation of the microtubules but also a loss in the organisaiton of their plus ends. The authors might wish to briefly comment/speculate on this?

P15 L14 The authors might want to speculate why the microtubule elongation phenotype is seen only for the corset microtubules and apparently not for axonemal microtubules?

P16 L1 Consideration of where the phosphorylation event occurs is an excellent discussion point, and one that would be enriched with some information on the localisation of CRK2, which is not touched on here. Have the authors carried out fractionation to test whether any of the CRK2 is cytoskeleton-associated? Fluorescence microscopy coupled to mild extraction might also be relevant and would go beyond published data. These should be noted as possible future experiments at least.

P16 L16 I think it would be more accurate to say that they have shown that a previously-documented control mechanism that utilises phosphorylation of beta-tubulin to regulate tubulin incorporation is also operating in trypanosomes, and coordinated by CRK2 instead of CDK1. Why CRK2 is used instead of CDK1 might be a good discussion point?

P16 L16+ Perhaps the authors might also want to speculate on whether there are implications for this control mechanism in the context of morphological changes that occur at different stages in the life cycle, for example the formation of mesocyclic cells, long epimastigotes, and so on?

P16 L26 For all cell lines, the authors should indicate how many separate clones were used for the different experiments, in order to control for biological variability. If only one clone was used, it is essential to demonstrate that the level of clone-to-clone variation is negligible.

P17 L5 The authors should probably explicitly mention that depletion of CRK2 and CYC13 protein by RNAi was demonstrated in their previous work, which explains why it is not shown here.

P17 L12 Was integration of the tagging construct at the endogenous locus confirmed by PCR analysis of genomic DNA?

P20 L6-10 The authors make extensive use of statistical significance tests, but state only that a t-test or a Chi-squared test was used. There are many different kinds of t-tests (two-sample t-test, Welch's t-test...) and it's not clear which they used. It's also unclear which test was applied when, which is important as the data are sometimes numerical and sometimes categorical, and the number of treatment categories varies. I would personally advocate removing all the significance tests from the figures (the results seem pretty clear-cut in most instances) but if the authors would prefer to keep them, then they need to provide more detail in the figure legends about what test was used and why.

P24 L2 The % fractions of the Input and pellet fractions need to be indicated (at the very least in the figure legend), so that the reader can get a sense of how much is being brought down.

The authors may wish to consider converting their red/green images into magenta/green ones - this has better contrast and is compatible with colour blindness (applies to immunuofluorescence, graphs, and schematics in Figures 1, 3, 4, 5, 6, 7).

Editorial points, minor comments: (in manuscript order)

Abstract & Summary

P2 L3 - delete "human". T. brucei is primarily a parasite of animals, not just humans.

P2 L3/4 - I think it would be more accurate to say "contains" or "features" instead of "is composed of" - there is more to the trypanosome cytoskeleton than just the corset microtubules.

P3 L1 perhaps rewrite as "...is a vector-borne parasite causing human and animal African trypanosomiasis..."? T.brucei is a parasite not just of humans.

Introduction

P4 L4 Delete the comma after "transport".

P5 L4 Delete "the" ("Clusters of alternating...").

Results

P6 L6 Maybe "more than five branches" instead? Calling the branches "posteriors" would suggest that normal posterior organisation is being duplicated in each branch.

P9 L9 Replace "assay" with "screen".

P10 L2 Maybe clarify that mutation to alanine and aspartate respectively are used for the two states? Not all readers will be familiar with this kind of approach.

P10 L9 Replace "more" with "higher".

P10 L10 Replace "less" with "lower".

Materials and Methods

P17 L16 Replace "transformed into" with "used to transform". Some more details on expression time, lysis method, and elution of bound protein would be informative.

P17 L19, P19 L13 Specify whether "NP-40" is Nonidet P-40 or Tergitol type NP-40. Replace "prepared by centrifugation" with "clarified by centrifugation".

P18 L6 could the authors briefly mention what the PNBM is doing? (for the benefit of general readers)

P18 L17 Maybe provide a little bit more detail on the trypsinisation - how much enzyme was used, and for how long?

P18 L27 Mention what counterion was used for setting the pH of the PIPES. NaOH?

P19 L11 Pham et al., 2019 should be cited with a number, and is not currently listed in the References section.

Figures/Figure legends

P22 L28 "causes" not "caused".

P23 L10 "impairs" not "impaired".

P24 L7 Maybe add a label to Figure 4 panel C to indicate that the phosphosites above the schematic are those identified following the in vitro assay, and the ones below the schematic are those identified by Ubaniak et al.?

P24 L10 "highlighted" not "outlined".

P24 L25 Figure 5 - I think it would be more accurate to say that the histogram shows the relative amounts of each tubulin construct present in the pellet and supernatant fractions.

P25 L6 I think it would be clearer to say that the beta-tubulin constructs were normalised against the loading control (TbPSA6) and then expressed relative to the first time point (0 hr).

Fig 6B - the labels say T351 but these should be S351.

Figure 6

P25 - The comments made on Figure 5 (above) also apply here.

Fig. S1

P26 L Maybe "data" instead of "spectrums"?

Brooke Morriswood

University of Würzburg

PLOS authors have the option to publish the peer review history of their article (what does this mean?). If published, this will include your full peer review and any attached files.

Reviewer #1: No

Reviewer #2: **Yes: **Daniel Abbühl and Philippe Bastin

Reviewer #3: **Yes: **Brooke Morriswood
---

## [Decision Letter · Decision Letter 1]

6 Mar 2023

Dear Dr. Li,

Thank you very much for submitting your manuscript "CRK2 controls cytoskeleton morphogenesis in Trypanosoma brucei by phosphorylating β-tubulin to regulate microtubule dynamics" for consideration at PLOS Pathogens. As with all papers reviewed by the journal, your manuscript was reviewed by members of the editorial board and by several independent reviewers. All are very supportive of publication but note that the referees suggest a couple of tiny tweaks to the text that I suggest you make to improve the submission. Thereafter, I will be pleased recommend 'acceptance' of the manuscript once those small changes are included. Further review will not be required. Thank you for the effort you have put in to improve this interesting work.

Sincerely,

Keith R. Matthews

Guest Editor

PLOS Pathogens

Margaret Phillips

Section Editor

PLOS Pathogens

Kasturi Haldar

Editor-in-Chief

PLOS Pathogens

orcid.org/0000-0001-5065-158X

Michael Malim

Editor-in-Chief

PLOS Pathogens

orcid.org/0000-0002-7699-2064

Reviewer Comments (if any, and for reference):

Reviewer's Responses to Questions

**Part I - Summary**

Reviewer #1: I have read the comments and changes made by the authors and they have addressed them very well. This is a nice piece of work that deserves to be published.

Reviewer #2: The authors have addressed all our requests and clarified or developed several points of importance. We believe that they have significantly improved their manuscript. There are a few small things to correct, but we don’t need to see the manuscript again.

Reviewer #3: The revised manuscript reads well and will clearly be a valuable addition to the literature. There is plenty of good quality data and some interesting ideas and implications that will undoubtedly stimulate further discussion. I'm grateful that the authors were so receptive to the (copious) feedback.

**Part II – Major Issues: Key Experiments Required for Acceptance**

Reviewer #1: no issues

Reviewer #2: (No Response)

Reviewer #3: None.

**Part III – Minor Issues: Editorial and Data Presentation Modifications**

Reviewer #1: no issues

Reviewer #2: One point for correction: after having read again the van der laan et al. paper (Cell reports 2019, reference 14) about the role of detyrosination of tubulin, we noticed that these authors also showed an impact on posterior elongation of the cytoskeleton. Since tubulin detyrosination is supposed to increase its stability, this result goes along the line proposed in the current manuscript and should be added in the discussion alongside the point about glutamylation (page 16, line 15).

Similarly, the sentence “only the polyglutamylation of microtubules has been functionally characterized (17,18)” of the introduction page 5, line 12) should be corrected since functional analysis is also presented for tubulin detyrosination, with the extension of the posterior end (and also some mitotic defects) in reference 14.

Finally, we noticed a few typos:

Page 5, lines 24 and 26 “polyglutamylation” (not polyglutamination)

Page 18, line 22 “might peak” (not might be peaked)

Page 20, line 6 “CRK2” (not CRk2)

Page 20, line 14 underlying mechanism *is* unknown

Page 21, line 21 “expression” (not expressing)

Daniel Abbühl and Philippe Bastin, Institut Pasteur, Paris

Reviewer #3: MINOR ISSUES:

None.

EDITORIAL POINTS:

Page references are taken from the page numbers listed on the manuscript text, not the PDF pages.

Author summary

P3 L3 Africa not African.

Results

P8 L27 Quantitative not quantitatively

P14 L22 "causes increased levels of" not "causes increased the level of"

There are a few very minor additional corrections that can be made to the text from a language perspective, but I will leave that to the copy editors.

Brooke Morriswood

University of Würzburg

PLOS authors have the option to publish the peer review history of their article (what does this mean?). If published, this will include your full peer review and any attached files.

Reviewer #1: No

Reviewer #2: **Yes: **Daniel Abbühl and Philippe Bastin, Institut Pasteur, Paris

Reviewer #3: **Yes: **Brooke Morriswood

Figure Files:

Data Requirements:

Reproducibility:

References:

---

## [Editor Report · Decision Letter 2]

8 Mar 2023

Dear Dr. Li,

We are pleased to inform you that your manuscript 'CRK2 controls cytoskeleton morphogenesis in Trypanosoma brucei by phosphorylating β-tubulin to regulate microtubule dynamics' has been provisionally accepted for publication in PLOS Pathogens.

Best regards,

Keith R. Matthews

Guest Editor

PLOS Pathogens

Margaret Phillips

Section Editor

PLOS Pathogens

Kasturi Haldar

Editor-in-Chief

PLOS Pathogens

orcid.org/0000-0001-5065-158X

Michael Malim

Editor-in-Chief

PLOS Pathogens

orcid.org/0000-0002-7699-2064
---

## [Editor Report · Acceptance letter]

17 Mar 2023

Dear Dr. Li,

We are delighted to inform you that your manuscript, "CRK2 controls cytoskeleton morphogenesis in Trypanosoma brucei by phosphorylating β-tubulin to regulate microtubule dynamics," has been formally accepted for publication in PLOS Pathogens.

Best regards,

Kasturi Haldar

Editor-in-Chief

PLOS Pathogens

orcid.org/0000-0001-5065-158X

Michael Malim

Editor-in-Chief

PLOS Pathogens

orcid.org/0000-0002-7699-2064